# A Variational Framework for Estimating Continuous Treatment Effects with Measurement Error

**Erdun Gao[1], Howard Bondell[1], Wei Huang[1] & Mingming Gong[1,2]**
[1]School of Mathematics and Statistics, The University of Melbourne
[2]Mohamed bin Zayed University of Artificial Intelligence
`erdun.gao@student.unimelb.edu.au`
`{howard.bondell,wei.huang,mingming.gong}@unimelb.edu.au`

## Abstract

Estimating treatment effects has numerous real-world applications in various fields, such as epidemiology and political science. While much attention has been devoted to addressing the challenge using fully observational data, there has been comparatively limited exploration of this issue in cases when the treatment is not directly observed. In this paper, we tackle this problem by developing a general variational framework, which is flexible to integrate with advanced neural network-based approaches, to identify the average dose-response function (ADRF) with the continuously valued error-contaminated treatment. Our approach begins with the formulation of a probabilistic data generation model, treating the unobserved treatment as a latent variable. In this model, we leverage a learnable density estimation neural network to derive its prior distribution conditioned on covariates. This module also doubles as a generalized propensity score estimator, effectively mitigating selection bias arising from observed confounding variables. Subsequently, we calculate the posterior distribution of the treatment, taking into account the observed measurement and outcome. To mitigate the impact of treatment error, we introduce a re-parametrized treatment value, replacing the error-affected one, to make more accurate predictions regarding the outcome. To demonstrate the adaptability of our framework, we incorporate two state-of-the-art ADRF estimation methods and rigorously assess its efficacy through extensive simulations and experiments using semi-synthetic data.

## 1 Introduction

Causality stands as a cornerstone concept across various scientific disciplines and serves as a powerful tool to unveil causal relationships among variables (Pearl, 2009; Peters et al., 2017; Hernan & Robins, 2023). One fundamental problem within causal inference research is identifying and estimating the causal effect of a treatment or policy, which can help guide rational decision-making in economics (Heckman, 2000), social science (Gangl, 2010), and public health research (Foster et al., 2011). In pursuit of this objective, the widely recognized gold standard involves the execution of randomized controlled trials (RCTs), which essentially entails implementing interventions. Although highly effective, this approach presents practical challenges due to its associated high costs, ethical considerations, and the complexities involved in ensuring compliance (Spirtes et al., 2000; Glymour et al., 2019). In response to this challenge, pursuing causal effect estimation using purely observational data, considered a more practical and applicable solution in certain contexts, has garnered a growing interest, both in academic and industrial fields (Imbens & Rubin, 2015; Chu et al., 2023).

Current causal effect estimation methods, such as nonparametric methods (Kennedy et al., 2017; Huber et al., 2020; Ai et al., 2021; Huang et al., 2022) and neural network-based methods (Shalit et al., 2017; Shi et al., 2019; Nie et al., 2020; Zhang et al., 2022; Wang et al., 2022) necessitate the accurate measurement of treatment data. Nevertheless, in numerous real-world scenarios, the observation of treatment and confounding variables may be subject to inaccuracies caused by diverse factors, including deliberate manipulation for confidentiality reasons and imprecise measurement

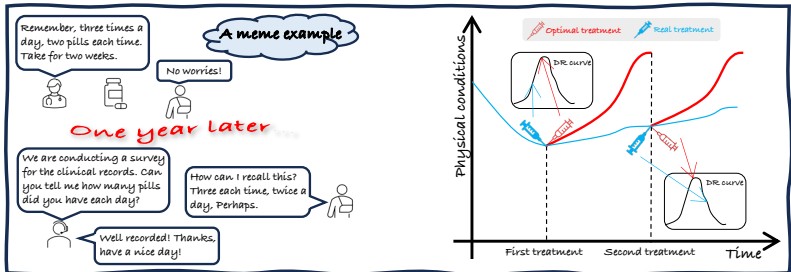

Figure 1: An illustrative example involving a meme demonstrates how a *error reporting* issue can lead to measurement errors. In the right subfigure, we depict the relationship between physical conditions and the treatment, with the DR curve denoting the dose-response curve.

techniques (Imai & Yamamoto, 2010; Schennach, 2016; Zhang et al., 2018; Grace et al., 2021). As an example, within the Epidemiologic Study Cohort data from the National Health and Nutrition Examination (NHANES-I), exceeding 75% of the variability observed in the fat intake data can be attributed to the presence of measurement errors (Carroll et al., 2006). Another noteworthy illustration can be found in the context of *error reporting*, as depicted in Fig. (1). In scenarios where we aim to evaluate the impact of a drug, patients might inadvertently misreport the true intake of the drug, especially if the record is completed after an extended period of treatment due to memory lapses.

**Challenges.** Even within the linear regression framework, a direct regression of the outcome on the imprecise measurement of covariates leads to attenuation error, which means an underestimation of the regression coefficient (Carroll et al., 2006; Zhu et al., 2022). In the context of treatment effect estimation, this attenuation error is manifested when estimating causal effects, indicating that neglecting measurement error can result in a biased model. Furthermore, methods designed for continuous treatment usually require (1) the estimation of the generalized propensity score (Hirano & Imbens, 2004) to extract the confounding variables in covariates and (2) learning the outcome based on the covariates and treatment. The impact of measurement error on the treatment variable is evident in both of these crucial steps.

To tackle these challenges, Huang & Zhang (2023) introduced a nonparametric approach for identifying and estimating the average dose-response function (ADRF). They proposed leveraging a weighted conditional expectation and nonparametrically estimating the associated weights. This entails maximizing a local generalized empirical likelihood while considering an expanding set of conditional moment equations, which are integrated into the deconvolution kernels (Delaigle & Hall, 2008). Subsequently, they constructed a deconvolution kernel estimator for the ADRF. It is worth noting, however, that nonparametric methods can be computationally intensive and face scalability issues when dealing with high-dimensional covariates (Bühlmann & Van De Geer, 2011). In another study (Zhu et al., 2022), this challenge was explored within the instrumental variable setting, where they devised a kernel-based nonparametric estimator (Singh et al., 2019). Nonetheless, their approach comes with a limitation as it necessitates the availability of at least two treatment measurements and an instrumental variable, a requirement that can be challenging to meet in practice. This constraint somewhat restricts the practical applicability of their method (Angrist et al., 1996).

**Contributions.** In this study, we present *a practical and general variational framework* (§ 3) designed for the estimation of treatment effect estimation in scenarios characterized by continuously valued error-contaminated treatments. Considering the data generating process, we focus on the identifiable model (Meister, 2006; Delaigle & Hall, 2016) where the measurement errors are assumed to be independent of covariates, treatment assignment, and potential outcomes (Assumption 2 in § 2). Our approach treats the unobservable treatment as a latent variable (§ 3.1) and leverages Bayesian techniques to approximate its posterior distribution based on the observed measurement variable and outcome. We then proceed to maximize the marginal likelihood of the observed variables to estimate the model parameters. This resulting variational framework (§ 3.2) is highly adaptable and can seamlessly integrate with various treatment effect estimation methods (Nie et al., 2020; Zhang et al., 2022) designed for error-free data. To evaluate the effectiveness of our proposed framework, we conducted extensive experiments across a wide range of settings, including multiple simulations (§ 4.1, § E.4) and three semi-synthetic datasets (§ 4.2). Our experiments involved comparisons with various state-of-the-art approaches (Ai et al., 2021; Huang & Zhang, 2023; Nie et al., 2020), providing robust evidence of the efficacy of our approach.

## 2 PROBLEM FORMULATION

**Notations.** We begin by presenting the notations and definitions used throughout this paper. In line with convention, we adhere to the convention of using capital letters for random variables (e.g., $X$), lowercase letters for their specific values (e.g., $x$), bold letters to represent sets of variables (e.g., $\mathbf{X}$), and their respective values (e.g., $\mathbf{x}$). The measurable spaces in which these variables are defined are represented using calligraphic letters (e.g., $\mathcal{X}$), with the exception of $\mathcal{P}$, which signifies a probability measure. Furthermore, we use $p_X(x)$ to represent the associated probability density function (pdf) for the random variable $X$.

In the context of the treatment effect problem, we denote $\boldsymbol{\mathcal{X}}$ as the $d$-dimensional space of covariates and $\mathcal{Y}$ as the outcome space. The observed treatment variable is represented by $T$, characterized by the pdf $p_T(t)$, and it operates within the domain $\mathcal{T} \subset \mathbb{R}$. For each unit with features $\mathbf{x} \in \boldsymbol{\mathcal{X}}$ and any treatment $t \in \mathcal{T}$, there exists a potential outcome $Y^t \in \mathcal{Y}$. In practice, each individual can only receive a single treatment level, denoted as $T$, leading to the observation of the corresponding outcome $Y$, represented as $Y := Y^t$. Here, we examine a scenario in which the treatment value is subject to classical measurement error. In this context, instead of directly observing $T$, we are provided with access to the measurement $S = T + U$, where $U$ denotes the measurement error.

**Task definition.** Suppose we have an independent and identically distributed (iid) sample with $N$ observations, denoted as $\{\mathbf{X}_i, S_i, Y_i\}_{i=1}^N$, which is drawn from the joint distribution of $(\mathbf{X}, S, Y)$. In this study, our objective is to estimate the unconditional ADRF, denoted as $\psi(t)$, defined as $\psi(t) := \mathbb{E}(Y^t)$. This represents the potential expected outcome that would have been observed under treatment level $t$.

Setting aside the data affected by errors, we begin with the error-free treatment scenario, saying that we have observed $T$. In order to establish the identifiability of the ADRF from observational data, we make the following assumptions consistently throughout this paper.

**Assumption 1.** *Assume that*

i. *[Unconfoundedness] for $\forall t \in \mathcal{T}$, given $\mathbf{X}$, $T$ is independent of $Y^t$, that is, $Y^t \perp\!\!\!\perp T | \mathbf{X} = \mathbf{x}$;*

ii. *[No Interference] The potential outcome for each individual $i$ remains independent of the treatment assigned to any other individuals. In other words, for any sets of treatments $\mathbf{T}_{-i}$ and $\mathbf{T}'_{-i}$, the potential outcome $Y_i^t(T_i, \mathbf{T}_{-i})$ for individual $i$ when given treatment $T_i$ and the rest of the individuals' treatments $\mathbf{T}_{-i} := (T_1, \ldots, T_{i-1}, T_{i+1}, \ldots, T_N)$, is equal to $Y_i^t(T_i, \mathbf{T}'_{-i})$;*

iii. *[Overlap] $\forall \mathbf{x} \in \boldsymbol{\mathcal{X}}$ such that $p_{\mathbf{X}}(\mathbf{x}) > 0$, we have $0 < p_{T|\mathbf{X}}(t|\mathbf{x}) < 1$ for each $t \in \mathcal{T}$;*

iv. *[Consistency] The observed outcome is the potential outcome that corresponds to the treatment level when the treatment is set to the observed exposure, saying $Y = Y^t$ if $T = t$ for $\forall t \in \mathcal{T}$.*

The **unconfoundedness**, also known as conditional exchangeability, asserts that *no hidden confounding* exists, thereby enabling the estimation of causal effects using observational data. The **overlap** assumption, also known as positivity, requires that each covariate combination has a non-zero probability density of receiving any treatment. These two assumptions are also referred to as the *strong ignorability* conditions (Rubin, 2005), commonly recognized within this field.

**Assumption 2.** *The measurement error $U$ is independent of $T$, $\mathbf{X}$, and $\{Y^t\}_{t \in \mathcal{T}}$, and its characteristic function is known up to certain parameters.*

To identify the error density, it is possible to relax the constraints of the known characteristic function by introducing replicated error-contaminated observations, as elaborated in (Delaigle et al., 2009). However, incorporating this method into our framework presents non-trivial challenges. More discussions can be found in (Huang & Zhang, 2023, Remark 1).

Under Assumption (1), the ADRF can be identified as

$$
\begin{aligned}
\psi(t) &= \mathbb{E}[\mathbb{E}(Y^t|\mathbf{X})] = \mathbb{E}[\mathbb{E}(Y^t|\mathbf{X}, T = t)] \quad \text{[with Assumption (i) and (ii)]} \\
&= \mathbb{E}[\mathbb{E}(Y|\mathbf{X}, T = t)] = \mathbb{E}[\mu(t, \mathbf{X})]. \quad \text{[with Assumption (iv)]}
\end{aligned}
\tag{1}
$$

One straightforward approach to estimate $\psi$ involves first obtaining an estimator $\hat{\mu}$ for $\mu$, and then constructing $\hat{\psi}(t)$ as the average of $\hat{\mu}(t, \mathbf{X}_i)$ over $N$ observations. To estimate $\psi$ efficiently, Hirano

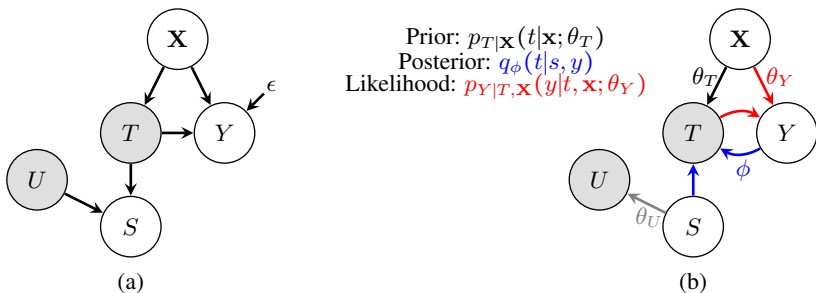

Figure 2: (a) Data generation process depicted by a DAG. The shaded circles denote latent variables. (b) **Learning an inference procedure**. we take $q_\phi(\cdot|s,y)$ to estimate the posterior distribution of $T$. The prior $p_{T|\mathbf{X}}(\cdot|\mathbf{x};\theta_T)$ is learned by a mixture network parameterized by $\theta_T$. The likelihood of outcome $p_{Y|\mathbf{X},T}(\cdot|\mathbf{x},t;\theta_Y)$ is estimated by a outcome prediction head parameterized by $\theta_Y$.

& Imbens (2004) proposed to replace the covariates by using the sufficient generalized propensity score (GPS) $\Pi = \pi(T,\mathbf{X})$, where $\pi(t,\mathbf{x}) = p_{T|\mathbf{X}}(t|\mathbf{x})$ is the conditional density of the treatment given the covariates. Then, we have

$$\psi(t) = \mathbb{E}[\mathbb{E}(Y|\pi(t,\mathbf{X}), T = t)]. \tag{2}$$

Compared to using all covariates, adjusting the GPS can efficiently remove biases associated with differences in covariates, focusing on extracting useful information for estimating $\psi$. In subsequent works (Shi et al., 2019; Nie et al., 2020), covariates are mapped into a feature space $\mathcal{Z}$, and the extracted features $\mathbf{Z}$ are employed to estimate the GPS $\pi(t,\mathbf{X})$ and $\mu(t,\mathbf{X})$. However, when the treatment is measured with errors, our observations are confined to $S$ instead of the ideal variable $T$. This poses a challenge in estimating $\psi(t)$ using Eq. (2), which we will address in the next section.

## 3 METHOODOLOGY

In this section, we introduce our proposed approach, which harnesses a variational framework to address the challenge of ADRF estimation with error-contaminated treatment. To begin, we outline a data generation process (§ 3.1) which elucidates the causal relationship between covariates, treatment, measurement error, and the outcome variable. We further decompose a conditional distribution derived from this model and propose to maximize the marginal likelihood of the observed variables as a means to solve this problem. The details are shown in the following subsections.

### 3.1 BAYESIAN METHODS FOR MEASUREMENT ERROR MODELS

We employ a probabilistic data generation model represented as a directed acyclic graph (DAG), as shown in Fig. (2 a), to illustrate the causal relationships among the variables under study. Here, $\mathbf{X}$ is the according covariates that would affect both the choice of treatment dosage $T$ and the outcome $Y$ as a confounder. According to this model, the data generation process can be represented as

$$\mathbf{X} \sim p_{\mathbf{X}}(\mathbf{x}); \quad T|\mathbf{X} \sim p_{T|\mathbf{X}}(t|\mathbf{x}); \quad S = T + U, \quad U \sim p_U(u);$$
$$Y = f(\mathbf{X}, T) + \epsilon, \quad \epsilon \sim \mathcal{N}(0, \sigma_\epsilon^2). \tag{3}$$

In this model, we assume that $\epsilon$ follows a normal distribution, although it can readily be substituted with any specified parametric family. As per this data generation process, we can factorize and explicitly express the conditional density of $p(t,s,y|\mathbf{x})$ as:

$$p_{T,S,Y|\mathbf{X}}(t,s,y|\mathbf{x}) = p_{Y|S,T,\mathbf{X}}(y|s,t,\mathbf{x})p_{S|T,\mathbf{X}}(s|t,\mathbf{x})p_{T|\mathbf{X}}(t|\mathbf{x})$$
$$= p_{Y|T,\mathbf{X}}(y|t,\mathbf{x})p_{S|T}(s|t)p_{T|\mathbf{X}}(t|\mathbf{x}). \quad \text{[with Assumption (2).]} \tag{4}$$

To parameterize the densities mentioned above, we utilize a neural network with parameters $\theta_Y$ to model the regression function $g(\cdot)$ for approximating $f(\cdot)$ and employ another neural network with parameters $\theta_T$ to represent the conditional density $p_{T|\mathbf{X}}(\cdot|\mathbf{x})$. Additionally, we use $\theta_U$ and $\theta_\epsilon$ to denote the density parameters of $p_U(\cdot)$ and $p_\epsilon(\cdot)$, respectively. Consequently, we can explicitly express the joint density of $(T, S, Y)$ w.r.t the covariates $\mathbf{X}$.

$$p(t,s,y|\mathbf{x};\Theta) = p_\epsilon(y - g(\mathbf{x},t;\theta_Y);\theta_\epsilon)p_U(s - t;\theta_U)p_{T|\mathbf{X}}(t|\mathbf{x};\theta_T), \quad \text{[change of variable.]} \tag{5}$$

where $\Theta := (\theta_T, \theta_Y, \theta_U, \theta_\epsilon)$ represent the concerned parameter set of the measurement error model. We employ Bayesian methods for parameter estimation, seeking to maximize the marginal log-likelihood of the observed variables $(S, Y, \mathbf{X})$ to estimate all these parameters:

$$L(\Theta) = \log \int p_\epsilon(Y - g(\mathbf{X}, t; \theta_Y); \theta_\epsilon) \times p_U(S - t; \theta_U) \times p_{T|\mathbf{X}}(t|\mathbf{X}; \theta_T)\mathrm{d}t, \tag{6}$$

The log-likelihood involves an integral over the latent variable $t$, which presents a challenge for analytical computation. Typically, parameter estimation and inference are accomplished using Markov chain Monte Carlo (MCMC) techniques (Crainiceanu et al., 2005) or, alternatively, the Monte Carlo EM algorithm (Ganguli et al., 2005). Nevertheless, these approaches come with significant computational costs, as they require computationally intensive MCMC procedures for posterior inference.

## 3.2 THE VARIATIONAL FRAMEWORK

To address this challenge, we draw inspiration from the variational auto-encoder framework, as outlined in prior works such as (Kingma & Welling, 2014; Burda et al., 2016; Tucker et al., 2018), which incorporates an additional neural network to facilitate variational inference. This approach can be viewed as an extension or relaxation of the conventional Expectation-Maximization algorithm, as described by (Dempster et al., 1977). In a more comprehensive explanation, we initially consider a *proposal* distribution for the unobserved treatment variable $T$ conditioned on $(S, Y)$, denoted as $T|(S, Y) \sim \mathcal{N}(m_\phi(S, Y), \sigma_\phi(S, Y))$, where the functions $m_\phi$ and $\sigma_\phi$ are parameterized by a neural network with parameters $\phi$. Let $q_\phi(t|s, y)$ represent this distribution. Leveraging Jensen's inequality, we derive a lower bound for the log-likelihood in Eq. (6):

$$\begin{aligned}
Q(\Theta, \phi) &= \log \int \frac{p(t, s, y|\mathbf{x}; \Theta)}{q_\phi(t|s, y)} q_\phi(t|s, y)\mathrm{d}t \\
&\geq \int \log \frac{p(t, s, y|\mathbf{x}; \Theta)}{q_\phi(t|s, y)} q_\phi(t|s, y)\mathrm{d}t = \mathbb{E}_{q_\phi(t|s,y)}\Big[ \log \frac{p(t, s, y|\mathbf{x}; \Theta)}{q_\phi(t|s, y)} \Big],
\end{aligned} \tag{7}$$

which is known as the evidence lower bound (ELBO) (Blei et al., 2017). Therefore, we can maximize this ELBO simultaneously over both the model parameters and the parameters $\phi$ of the proposal distribution. Moreover, the ELBO can be factorized as

$$\begin{aligned}
Q(\Theta, \phi) &= \mathbb{E}_{q_\phi(t|s,y)}\Big[ \log \frac{p_{Y|T,\mathbf{X}}(y|t, \mathbf{x})p_{S|T}(s|t)p_{T|\mathbf{X}}(t|\mathbf{x})}{q_\phi(t|s, y)} \Big] \\
&= \mathbb{E}_{q_\phi(t|s,y)}[\log p_{Y|T,\mathbf{X}}(y|t, \mathbf{x}) + \log p_{S|T}(s|t)] + \mathbb{E}_{q_\phi(t|s,y)}\Big[ \log \frac{p_{T|\mathbf{X}}(t|\mathbf{x})}{q_\phi(t|s, y)} \Big] \\
&= \mathbb{E}_{q_\phi(t|s,y)}[\log p_\epsilon(y - g(\mathbf{x}, t; \theta_Y); \theta_\epsilon) + \log p_U(s - t; \theta_U)] - \mathrm{KL}[q_\phi(t|s, y)\|p_{T|\mathbf{X}}(t|\mathbf{x}; \theta_T)].
\end{aligned} \tag{8}$$

We can then optimize Eq. (8) in lieu of Eq. (6), which does not have a closed-form expression and requires expensive MCMC to evaluate. The overall structure is illustrated in Fig. (2 b). However, we are still left with the functional form of the outcome prediction module $g(\cdot)$ and prior $p_{T|\mathbf{X}}(\cdot)$, both of which are to be chosen before optimization Eq. (6) w.r.t. $\Theta$.

**Prior estimator $p_{T|\mathbf{X}}(t|\mathbf{x}; \theta_T)$:** Recall that the initial step of this conditional density estimator, which involves the mapping of covariates $\mathbf{X}$ into a feature space, yielding the corresponding feature vector $\mathbf{Z}$. This feature extraction process is executed by a simple network, similar to the approach employed in prior studies such as Nie et al. (2020); Zhang et al. (2022). Following this feature extraction stage, we employ a mixture density network, as originally introduced by Bishop (1994), to estimate the conditional density. Alternatively, various other techniques can be utilized for this purpose, including the kernel mixture network (Ambrogioni et al., 2017) and normalizing flows (Trippe & Turner, 2018). It is worth noting that our methodology aligns with the approach utilized in (Hirano & Imbens, 2004) by using the GPS instead of the raw covariates to predict the outcome. This choice is motivated by the equivalence between conditioning on the features is equivalent to conditioning on the propensity score itself (Shi et al., 2019). Consequently, this estimator also plays a dual role in obtaining the GPS.

**Outcome prediction module $g(\mathbf{x}, t; \theta_Y)$:** As previously explored in Shalit et al. (2017), the role of the treatment variable $T$ distinctly differs from that of the covariates vector $\mathbf{X}$, potentially leading

Figure 3: Comparing AMSE (↓), APE (↓) and Runtime (↓) of our proposed methods VF-VCNet and VF-ATTNet and baseline methods on 1-d covariate simulated Data. We use the MLP Function from covariates to generate treatment. Columns: Varying Observation Counts (First Three) and Varying Measurement Error Standard Deviation (Last Two).

to the attenuation of $T$'s impact when considered within the context of high-dimensional features. To tackle this challenge, Schwab et al. (2019) introduces a novel approach named dose-response network (DRNet) that advocates the use of separate prediction heads for $T$ corresponding to different value intervals. In Nie et al. (2020), a varying coefficient neural network (VCNet) is designed to simultaneously emphasize the influence of treatment while preserving the continuity of the ADRF. Furthermore, Zhang et al. (2022) explores the Transformer architecture (Vaswani et al., 2017) and proposes an attention neural network (ATTNet[1]) to capture the intricate interactions between the treatment and covariates, offering a parameter-efficient and robust solution to address shifts in treatment effects. Within our methodology, we have the flexibility to incorporate both VCNet and ATTNet, referred to as VF-VCNet and VF-ATTNet, respectively, in the subsequent discussions as viable choices for the outcome prediction module. Additionally, our framework is adaptable to incorporate forthcoming advanced models. Nevertheless, integrating DRNet, with its inherent discontinuities, into our framework may present challenges.

## 3.3 TRAINING AND INFERENCE

According to Eq. (8), we can optimize all parameters by minimizing the following loss:

$$\mathcal{L}(\Theta, \phi) = -\mathbb{E}_{q_\phi(t|s,y)}\big[\log p_\epsilon(y - g(\mathbf{x}, t)) + \alpha \log p_U(s - t)\big] + \beta \text{KL}\big[q_\phi(t|s,y)\|p_{T|\mathbf{X}}(t|\mathbf{x})\big]. \quad (9)$$

The value of $\beta$, which is a hyper-parameter followed the way in (Higgins et al., 2016), is used in the computation of Kullback–Leibler (KL) divergence between the posterior and prior of $T$. Additionally, we have observed improved performance by introducing an external parameter $\alpha$ to regulate the relative weight of the measurement error loss in our method. In our method, we exclusively employ the vanilla Variational Autoencoder (VAE) (Kingma & Welling, 2014) to address this issue. However, it is worth noting that VAE relies on a strong assumption that the true conditional distribution of $T$ can be effectively approximated by a simple proposal distribution. For those seeking to further reduce the Monte Carlo (MC) variance, advanced techniques such as Importance Weighted Autoencoder (IWAE) (Burda et al., 2016) and Doubly Reparameterized Gradient (DReG) (Tucker et al., 2018) can also be employed.

Let $\hat{\mu}$ and $\hat{\pi}$ represent the optimal solutions derived from the optimization problem (9). Once we have obtained $\hat{\mu}$, we can estimate $\psi(\cdot)$ as $\hat{\psi}(\cdot) = \frac{1}{N} \sum_{i=1}^{N} \hat{\mu}(\cdot, \mathbf{X}_i)$.

## 4 EXPERIMENTAL RESULTS

We report the empirical results to verify the effectiveness of our variational framework in accurately estimating continuous treatment effects in the presence of measurement errors on the treatment variable, across multiple synthetic datasets and three semi-synthetic datasets, specifically IHDP (Hill, 2011), News (Newman, 2008), and TCGA (Weinstein et al., 2013).

**Baselines.** We compare our methods named VF-VCNet and VF-ATTNet against two nonparametric methods named MC-Naive and MC-SIMEXLC (Huang & Zhang, 2023), which identifies the ADRF by a weighted conditional expectation. Another way to deal with the measurement error problem is to

---

[1]We name it ATTNet instead of TransTEE used in (Zhang et al., 2022) due to some specific modifications in this method. More details are in Appendix D.1.

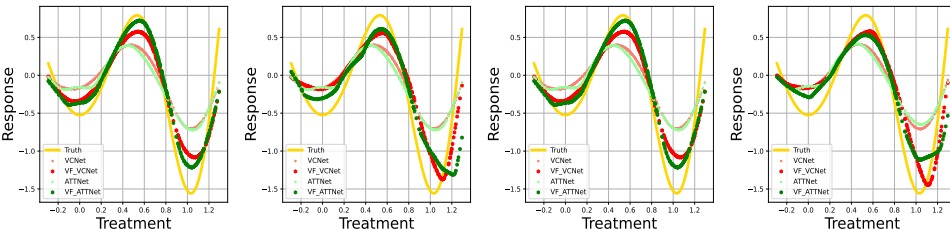

Figure 4: Estimated ADRF for VCNet, ATTNet, VF-VCNet, and VF-ATTNet is provided for the testing set of four simulation datasets. Each dataset exhibits unique characteristics in terms of treatment and error distributions. Specifically, from left to right, these datasets correspond to the following treatment distribution-error distribution pairs: normal-normal, normal-laplace, uniform-normal, and uniform-laplace. The ground truth is shown in the yellow solid line.

directly ignore the measurement error and treat the problem as a standard ADRF estimation problem. When the variance of $U$ is properly small, this method can have a reasonably good performance. For this case, we consider four advanced methods including TARNet (Shalit et al., 2017), DRNet (Schwab et al., 2019), VCNet (Nie et al., 2020), and ATTNet (Zhang et al., 2022) as baselines. To ensure a fair comparison, we employ a consistent prior estimator for all methods. Specifically, we utilize a Gaussian density estimator when dealing with normal conditional priors, and opt for a mixture network in non-Gaussian scenarios. Detailed implementations are in Appendix D.1.

**Metrics.** In this paper, we present our findings using two key evaluation metrics. Firstly, we employ the widely recognized metric known as the average mean squared error (AMSE) (Bica et al., 2020; Zhang et al., 2022; Wang et al., 2022). This metric quantifies the proximity of our predicted outcomes to the true potential outcomes when considering the covariates and the corresponding treatment. Additionally, once we have obtained the ADRF curve, determining the optimal treatment becomes paramount. To get this, we introduce the average policy error (APE) (Bica et al., 2020), which measures the absolute discrepancy between our estimated optimal treatment and the actual optimal treatment. All results presented follow a format of mean ± standard deviation, which is calculated on the test set from an average of over 10 independent random repetitions. For more comprehensive information, please refer to Appendix D.3.

## 4.1 SYTHETIC DATA.

**Data generation.** Given the scarcity of access to true ADRF in real-world data, conventional approaches for estimating treatment effects often resort to using synthetic or semi-synthetic datasets for empirical assessments, as observed in prior research (Bica et al., 2020; Nie et al., 2020). In our study, we conduct two simulations: one with a one-dimensional covariate and the other with six randomly generated covariates. These simulation datasets are generated using a data-generating mechanism that utilizes the randomly generated covariates to determine the treatment. Specifically, we employed two kinds of functions, multilayer perceptron (MLP) and multiple index model (MiM), for the one-dimensional case, and a function similar to the one described in (Nie et al., 2020) for the six-dimensional case. Then, we use a complex outcome-determination process given the covariate(s) and the treatment. We also choose different prior for $T$ and distributions for $U$. For further dataset details, please refer to Appendix B.

**Results:** In Fig. (3), across all settings for VF-VCNet and VF-ATTNet including varying observation numbers and varying error variance can show consistently the best performance. Among the baseline methods, MC-SIMEXLS (Huang & Zhang, 2023) consistently achieves the third-best performance due to its incorporation of a deconvolution kernel estimator, which effectively accounts for measurement error. However, it's worth noting that MC-SIMEXLC, while exhibiting strong performance, incurs the longest execution time, which results from its utilization of bootstrap techniques to further enhance its performance, which is a computationally intensive process. In the context of six-dimensional covariates, we have excluded MC-Naive and MC-SIMEXLC from our baseline comparisons. It is challenging to adapt their methodologies to high-dimensional data since such extensions are highly sensitive to the hyperparameters selection and require a prohibitively long runtime to execute these methods. In Fig. (4), we visualize the estimated ADRF for both our methods

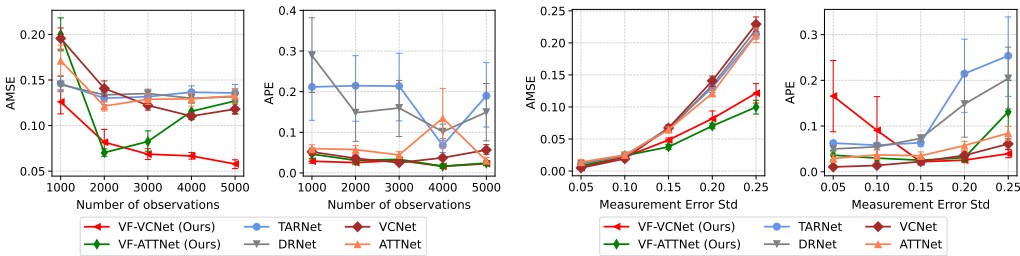

Figure 5: Comparing AMSE and APE for VF-VCNet, VF-ATTNet, and baselines on 6-D covariate Simulated Data. We take the normal measurement error. Columns: Varying Observation Counts (First Two) and Varying Measurement Error Standard Deviation (Last Two).

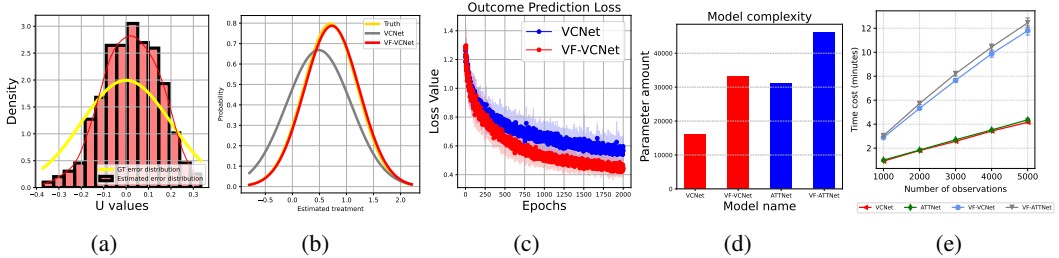

Figure 6: Results of the ablations studies. (a) Estimated ME of VF-VCNet on the testing data; (b) Comparative assessment of estimated conditional density; (c) Evaluation of outcome prediction loss; (d) Number of parameters for different models; (e) Comparative assessment of running times.

and their respective base methods, which highlights the enhanced accuracy achieved by our methods in the estimation process. We provide a quantitative analysis of these results in Fig. (5), further underscoring the superior performance of our methods across all tested configurations.

### 4.1.1 ABLATION STUDIES

**Estimation measurement, conditional density, and the outcome.** In Fig. (6 a,b,c), we illustrate the efficacy of our method using an example that compares the estimated results of VF-VCNet with those of VCNet. This comparison demonstrates the superior performance of VF-VCNet in estimating both the conditional density $p_{T|\mathbf{X}}(t|\mathbf{x})$ and the regression function $g(\cdot)$ in $p_{Y|T,\mathbf{X}}(y|t,\mathbf{x})$, as evidenced by its lower regression loss. The key advantage of VF-VCNet lies in its ability to effectively separate the measurement error, thereby enhancing its estimation capabilities.

**Model complexities.** We offer a comprehensive assessment of model complexities and computational times for our methods in comparison to their base counterparts in Fig. (6 d,e). Integrating our variational framework introduces an additional posterior estimation module $q_\phi(t|s,y)$, which leads to an expansion in model parameters and an increase in convergence time. However, these increments in model complexity and execution time remain well within acceptable limits.

### 4.2 SEMI-SYTHETIC DATA

**Datasets.** *IHDP-Continuous:* The semi-synthetic IHDP dataset from (Hill, 2011) contains binary treatments with 747 observations on 25 covariates. We modify this dataset to align with the continuous dosage context by adjusting the treatment assignment and outcome function. *News-continuous:* The News dataset consists of 3000 randomly sampled news items from the NY Times corpus (Newman, 2008) and was originally introduced as a benchmark for binary treatment settings. To adapt it to the continuous treatment context, we generate continuously-valued treatments and corresponding outcomes. *TCGA-continuous:* The TCGA dataset encompasses gene expression measurements for cancer patients (Weinstein et al., 2013). There are in total 9659 samples for which we used the measurements from the 4000 most variable genes. We generate continuous treatment and outcomes to adapt to our task. More details on constructing the datasets can be seen in Appendix B.

Table 1: Performance of ADRF on three datasets. Metrics are reported as Mean ± Std.

| Method | IHDP | | News | | TCGA | |
|---|---|---|---|---|---|---|
| | AMSE | APE | AMSE | APE | AMSE | APE |
| TARNet | 0.100 ± 0.045 | 0.073 ± 0.094 | 0.189 ± 0.144 | 0.078 ± 0.094 | 0.042 ± 0.007 | 0.136 ± 0.069 |
| DRNet | 0.090 ± 0.021 | 0.068 ± 0.007 | 0.186 ± 0.121 | 0.058 ± 0.007 | 0.047 ± 0.007 | 0.182 ± 0.040 |
| ATTNet | 0.088 ± 0.043 | 0.088 ± 0.043 | 0.296 ± 0.171 | 0.000 ± 0.000 | 0.056 ± 0.019 | 0.076 ± 0.050 |
| VCNet | 0.129 ± 0.037 | 0.104 ± 0.080 | 1.141 ± 0.883 | 0.688 ± 0.439 | 0.046 ± 0.010 | 0.067 ± 0.045 |
| VF-ATTNet | 0.099 ± 0.049 | 0.044 ± 0.049 | 0.589 ± 0.386 | 0.000 ± 0.000 | 0.042 ± 0.010 | 0.057 ± 0.037 |
| VF-VCNet | 0.068 ± 0.027 | 0.057 ± 0.044 | 0.714 ± 0.441 | 0.521 ± 0.436 | 0.019 ± 0.014 | 0.036 ± 0.017 |

**Semi-Sythetic Data.** In Table 1, we compare our approach with the baseline methods TARNet, DRNet, VCNet, and ATTNet. In our analysis of the IHDP and TCGA datasets, we note that two extended methods, VF-VCNet and VF-ATTNet, consistently outperform the baseline models by a substantial margin. Shifting our focus to the News dataset, our approach VF-VCNet demonstrates an improvement in both AMSE and APE compared to the base model VCNet. However, it's worth mentioning that the performance of VF-ATTNet diminishes in this context due to (1) the complexity of the conditional distribution $p(T|\mathbf{X})$, and (2) the limited observations. We have two more datasets in Appendix E.4 to show the performance of our proposed with high-dimensional covariates. More detailed analysis can be found in Appendix E.3.

## 5 RELATED WORK

**Treatment effect estimation.** Instead of conducting RCTs, (Rosenbaum & Rubin, 1983) introduced propensity score matching as a method to balance covariates between treated and control groups. (Hirano & Imbens, 2004) extends the propensity score to the GPS to deal with continuous treatment. In recent years, there has been a growing interest in leveraging neural networks for treatment effect estimation, with various approaches emerging. One pioneering approach, introduced by (Shalit et al., 2017), involves extracting a feature representation for covariates and employing separate neural heads for different treatment groups. To extend the binary case to continuous treatment scenarios, DRNet (Shi et al., 2019) utilizes multi-head neural networks. Bica et al. (2020) designs the SCIGAN approach, which harnesses generative adversarial networks to generate counterfactual outcomes for continuous treatments. VCNet (Nie et al., 2020) takes a novel approach by incorporating varying coefficient networks, while ATTNet (Zhang et al., 2022) leverages attention mechanisms to model the outcome regression function. More detailed related works can be found in Appendix A.

## 6 DISCUSSION

**Conclusion:** In this paper, we introduce a general variational framework to estimate the average dose-response function in scenarios where the treatment is not directly observed and is subject to measurement errors. We treat the unobserved treatment as a latent variable, employing a proposal distribution for it. Then, we maximize the ELBO of the log-likelihood to estimate the model parameters. This framework offers a flexible and effective approach, showcased through simulations and semi-synthetic data experiments. It holds significant potential for applications in fields such as epidemiology and political science, where causal effects estimation is crucial but challenging due to measurement errors.

**Limitaions.** Our methods exhibit two primary limitations. Firstly, they depend on an increased number of parameters, necessitating a larger dataset for effective learning. Secondly, our approaches encounter a challenge in the absence of treatment supervision for learning the GPS, particularly when dealing with intricate underlying conditional distributions $p(T|\mathbf{X})$.

**Future works:** Our current method addresses independent measurement errors, but in reality, these errors can also be impacted by covariates. We should explore extending our framework to handle these complexities. Additionally, a crucial future direction is considering hidden confounders beyond the *ignorability* assumption.

## 7 ACKNOWLEDGEMENTS

The authors would like to thank Kenyon Ng for engaging in discussions regarding the method design and for his valuable contributions to refining the paper. EG is supported by an Australian Government Research Training Program (RTP) Scholarship. This research was undertaken using the LIEF HPC-GPGPU Facility hosted at the University of Melbourne. This Facility was established with the assistance of LIEF Grant LE170100200. MG was supported by ARC DE210101624. HB was supported by ARC FT190100374.

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

# Appendix

## Table of Contents

# A    ADDITIONAL RELATED WORKS

## A.1    MEASUREMENT ERROR MODELS.

There are mainly two lines to deal with the measurement error problem. The first approach, rooted in Bayesian methods (Crainiceanu et al., 2005; Ganguli et al., 2005; Hu et al., 2022), involves making assumptions about underlying probability distributions. This allows for the derivation of the marginal likelihood of observable variables and the subsequent maximization of likelihood when estimating model parameters. The second approach, known as nonparametric methods, takes inspiration from established techniques for the error-free scenario and endeavors to rectify biases introduced by measurement errors. For instance, the deconvolution approach (Delaigle & Hall, 2008; Delaigle et al., 2009) builds upon local smoothing estimators commonly employed in nonparametric regression under ideal conditions. Meanwhile, regression spline methods (Jiang et al., 2023) estimate the posterior distribution of latent variables and compute an unbiased score function. Another noteworthy technique, the SIMEX approach (Staudenmayer & Ruppert, 2004), extends from arbitrary nonparametric regression methods designed for error-free situations.

## A.2    NONPRAMETRIC METHODS FOR TREATMENT EFFECT ESTIMATION

Another line for estimating treatment effects involves the utilization of nonparametric methods. Galvao & Wang (2015) identified the ADRF using an unconditional weighted expectation, where the weighting function is derived from the ratio of two conditional densities of the treatment variable. They estimated these two conditional densities separately and subsequently constructed the ADRF estimator. Kennedy et al. (2017) developed a doubly robust estimator for the ADRF by conducting a regression analysis on a doubly robust mapping with respect to the treatment variable. Ai et al. (2021) employed a method to identify the ADRF through a weighted conditional expectation regarding the treatment variable, with the weighting function being defined as the ratio of the treatment variable's marginal density to its conditional density given the confounders.

## A.3    MORE CASES FOR TREATMENT EFFECT ESTIMATION

Additionally, Wang & Blei (2019) challenged the *ignorability* assumption to allow unobserved confounder and introduced the *deconfounder* method to amalgamate unsupervised machine learning and predictive model checking for causal inference in settings involving multiple causes. This method infers a latent variable to act as a substitute for unobserved confounders, subsequently employing this substitute in causal inference procedures. Mahajan (2006) examined the binary treatment effect in the presence of misclassified observed treatment indicators, while Battistin & Chesher (2014) explored how measurement errors in confounders impact the analysis of treatment effects. Shu & Yi (2019) investigated the influence of measurement errors on both continuous and discrete outcome variables and devised consistent estimation methods suitable for practical situations involving either validation data or replicates.

## B  DATASET

### B.1  GENERAL SETTING

In Section 4, our experimental setting includes two main categories, namely, *Varying the Number of Observations* and *Varying the Variance of Measurement Error*. To ensure fair comparisons, we consistently generate a dataset comprising 10000 observations for each trial. Subsequently, we extract the specified number of observations from this dataset while reserving a constant set of 200 observations for testing the trained model. As outlined earlier, our methodology entails conducting 10 experiments for each configuration, meaning we randomly generate 10 different sample. For IHDP and news, we randomly split into a training set (67%) and a testing set (33%).

### B.2  SIMULATIONS WITH ONE-DIMENSIONAL COVARIATES

To compare with MC-Naive (Delaigle & Hall, 2008) and MC-SIMEXLC (Huang & Zhang, 2023), we follow a similar way to generate all the simulated data, while making the conditional density of treatment $p(t|\mathbf{x})$ more complex. We first generate the covariate $X \in \mathbb{R}$ drawn from a uniform distribution between $0$ and $1$. Then, when generating the continuous dosages, we adopted two functions named MLP and MIM, respectively. MLP model is a randomly initialized MLP with one hidden layer of size $100$ and sigmoid activation. MIM model is $f(x) = \sum_{m=1}^{3} h_m(\theta_{jm}x)$, where $h_1 = \tanh$, $h_2 = \cos$, $h_3 = \sin$, and each $\theta_{jm}$ is drawn uniformly from range $[-2, -0.5] \cup [0.5, 2]$. Similarly, we rescale the output to an interval by $t = (1 + \exp(-\tilde{t}))^{-1}$ and add a random noise sampled from a Gaussian distribution $\mathcal{N}(0, 0.25)$ to obtain the treatment. Then, we add the measurement error to get the error-contaminated measurement $S$. Finally, we use a similar way as in (Nie et al., 2020) to generate the outcome:

$$y|x,s = \cos(2\pi(s - 0.5))\left(s^2 + \frac{4x^3}{1 + 2x^2}\sin(x)\right) + \mathcal{N}(0, 0.25). \tag{10}$$

### B.3  SIMULATIONS WITH SIX-DIMENSIONAL COVARIATES

We generate synthetic data similar to (Nie et al., 2020). With covariates $\mathbf{x} \in \mathbb{R}^6$ all drawn from a uniform distribution between $0$ and $1$, we generate the continuous dosages and outcomes as follows,

$$\tilde{t}|\mathbf{x} = \frac{10\sin(\max(x_1, x_2, x_3)) + \max(x_3, x_4, x_5)^3}{1 + (x_1 + x_5)^2} + \sin(0.5x_3)(1 + \exp(x_4 - 0.5x_3)) + \tag{11}$$
$$x_3^2 + 2\sin(x_4) + 2x_5 - 6.5,$$
$$s|t, u = t + \mathcal{D}(U), \tag{12}$$
$$y|\mathbf{x}, s = \cos(2\pi(s - 0.5))\left(s^2 + \frac{4\max(x_1, x_6)^3}{1 + 2x_3^2}\sin(x_4)\right) + \mathcal{N}(0, 0.25), \tag{13}$$

where $t = (1 + \exp(-\tilde{t}))^{-1} + \mathcal{N}(0, 0.25)$ and $\mathcal{D}(U)$ means that sampling an observation from the distribution of measurement error. Notice that $\pi(t \mid x)$ only depends on $x_1, x_2, x_3, x_4, x_5$ while $Q(t, x)$ only depends on $x_1, x_3, x_4, x_6$. As discussed in Shi et al. (2019), this allows us to observe the improvement using all these methods when noise covariates exist.

### B.4  IHDP CONTINUOUS

The IHDP dataset contains 25 covariates with binary treatments and continuous outcomes Hill (2011). Disregarding the treatments and outcomes, we use the covariates to generate continuous dosages and treatments to test our method. We follow the data generating procedure of (Nie et al., 2020). The

dosages are generated according to,

$$\tilde{t}|\mathbf{x} = \frac{x_1}{1+x_2} + \frac{\max(x_3, x_5, x_6)}{0.2 + \min(x_3, x_5, x_6)} + 2\tanh\left(5\frac{\sum_{i \in I} x_i - c_2}{|I|}\right) - 2, \tag{14}$$

$$s|t, u = t + \mathcal{D}(U), \tag{15}$$

$$y|\mathbf{x}, s = \frac{\sin(1.5\pi s)}{2-s}\left(\tanh\left(5\frac{\sum_{i \in J} x_i - c_1}{|J|}\right) + \exp\left(\frac{0.2(x_1 - x_6)}{0.5 + \min(x_2, x_3, x_5)}\right)\right) + \mathcal{N}(0, 0.25), \tag{16}$$

$$c_1 = \mathbb{E}_{p(\mathbf{x})}\left[\frac{\sum_{i \in J} x_i}{|J|}\right], \tag{17}$$

$$c_2 = \mathbb{E}_{p(\mathbf{x})}\left[\frac{\sum_{i \in I} x_i}{|I|}\right], \tag{18}$$

where $t = (1 + \exp(-\tilde{t}))^{-1} + \mathcal{N}(0, 0.25)$, $\mathcal{D}(U)$ means that sampling an observation from the distribution of measurement error, and $I = \{16, 17, 18, 19, 20, 21, 22, 23, 24, 25\}$, and $J = \{4, 7, 8, 9, 10, 11, 12, 13, 14, 15\}$.

## B.5 NEWS CONTINUOUS

This dataset contains words sampled from 5000 news articles Newman (2008). The covariates are word counts. We generated continuous dosage and outcomes by following the data generation method listed in Bica et al. (2020); Nie et al. (2020). We first sample three vectors $\mathbf{v}'_i \sim \mathcal{N}(0, 1)$, with $\mathbf{v}_i = \mathbf{v}'_i / \|\mathbf{v}'_i\|_2$ for $i = 1, 2, 3$. The dosages are drawn from a distribution $t \sim \text{Beta}(\alpha, t_w)$, where we set $\alpha = 2$ and,

$$t_w = \max\left(1, \left|\frac{2\mathbf{x}^T\mathbf{v}_2}{\mathbf{x}^T\mathbf{v}_1}\right|\right). \tag{19}$$

Notice that in this case, the conditional distribution $p(t|\mathbf{x})$ follows a beta distribution, which can be approximated by a Gaussian mixture model. Then, we generated the error-contaminated observable $S$ by

$$s|t_w, u = t_w + \mathcal{D}(U), \tag{20}$$

where $\mathcal{D}(U)$ means sampling an observation from the distribution of measurement error. Now we sample the outcomes according to,

$$y' = \exp\left(\left|\frac{\mathbf{x}^T\mathbf{v}_2}{\mathbf{x}^T\mathbf{v}_1}\right| - 0.3\right) \tag{21}$$

$$y = 2\left(\max(-2, \min(2, y') + 20\mathbf{x}^T\mathbf{v}_3 * (4(s-0.5)^2) * \sin\left(\frac{\pi s}{2}\right)\right) + \mathcal{N}(0, 0.25). \tag{22}$$

## B.6 TCGA CONTINUOUS

We get the dataset from Weinstein et al. (2013), which includes 9659 observations and each observation has 4000 covariates. To generate the treatment from covariates, we first randomly choose 100 covariates and use a two-layer MLP of size $100 \times 100 \times 1$. Similarly, we rescale the output to an interval by $t = (1 + \exp(-\tilde{t}))^{-1}$ and add a random noise sampled from a Gaussian distribution $\mathcal{N}(0, 0.25)$ to obtain the treatment. Then, we add the measurement error to get the error-contaminated measurement $S$.

Then, to generate the outcome, we follow the same way in Bica et al. (2020); Wang et al. (2022). We first generate a set of parameters $\mathbf{v}_1, \mathbf{v}_2, \mathbf{v}_3$. For each run of the experiment, these parameters are sampled randomly by sampling a vector, $\mathbf{u}$, from $\mathcal{N}(\mathbf{0}, \mathbf{1})$ and then setting $\mathbf{v} = \mathbf{u}/\|\mathbf{u}\|$ where $\|\cdot\|$ is Euclidean norm. Then we generate the outcome given the covariates and the treatment in the following way.

$$y = f(\mathbf{x}, s) = ((\mathbf{v}_1^2)^T\mathbf{x} + \sin(\pi(\frac{\mathbf{v}_2^{2T}\mathbf{x}}{\mathbf{v}_3^{2T}\mathbf{x}})s)) + \mathcal{N}(0, 0.25). \tag{23}$$

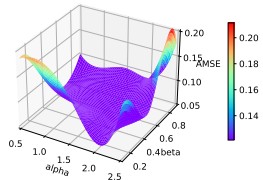

Figure 7: The sensitivity of our method to hyper-parameters $\alpha$ and $\beta$.

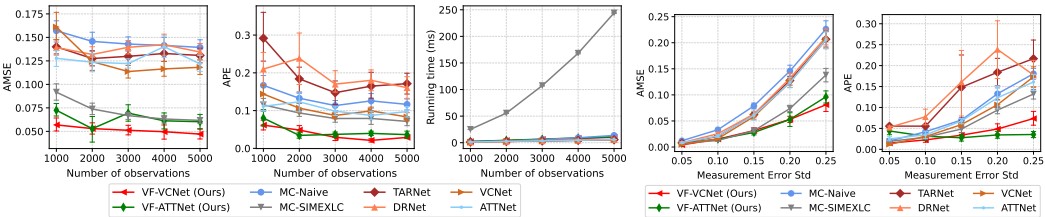

Figure 8: Comparing AMSE ($\downarrow$), APE ($\downarrow$), and Runtime ($\downarrow$) for VF-VCNet, VF-ATTNet, and baselines on 1-d covariate simulated Data. MIM Function; Columns: Varying Observation Counts (First Three) and Varying Measurement Error Standard Deviation (Last Two).

## C    ADDITIONAL EXPERIMENTS

In this subsection, we provide more experimental results on the simulated datasets.

### C.1    PARAMETER SENSITIVITY.

Fig. (7) illustrates the sensitivity of our method to hyper-parameters $\alpha$ and $\beta$. Specifically, we vary $\alpha$ over the range $\{0.5, 1.0, 1.5, 2.0, 2.5\}$ and $\beta$ over the range $\{0.2, 0.4, 0.6, 0.8, 1.0\}$. It can be observed that our method achieves competitive performance robustly across a wide range of hyper-parameter values. In all experiments, we set $\alpha$ as $1.5$ and $\beta$ as $0.5$.

### C.2    MORE COMPARISONS ON 1-D SIMULATED DATA.

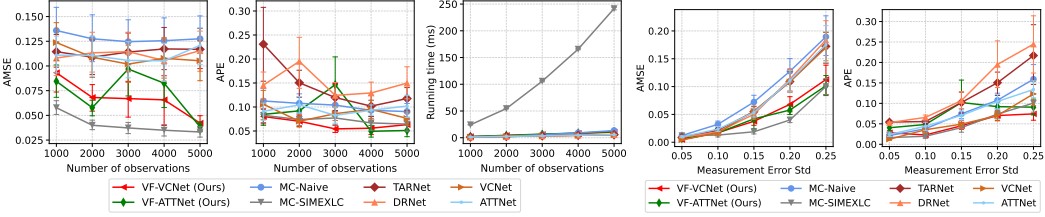

Figure 9: Comparing AMSE ($\downarrow$), APE ($\downarrow$), and Runtime ($\downarrow$) for VF-VCNet, VF-ATTNet, and baselines on 1-D covariate simulated Data. MLP function between $T$ and $X$. Columns: Varying Observation Counts (First Three) and Varying Measurement Error Standard Deviation (Last Two). The conditional density type of $T$: Normal; The noise type of measurement error $U$: Laplace.

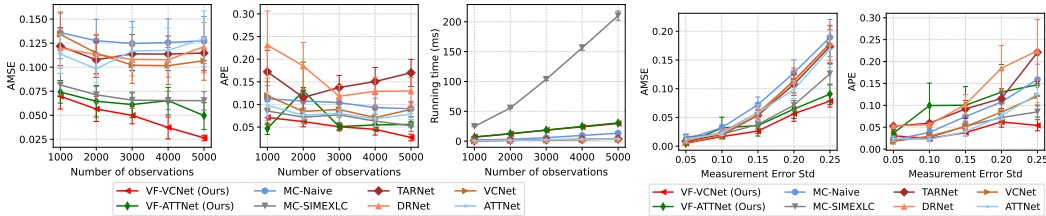

Figure 10: Comparing AMSE (↓), APE (↓), and Runtime (↓) for VF-VCNet, VF-ATTNet, and baselines on 1-D covariate simulated Data. MLP function between $T$ and $X$. Columns: Varying Observation Counts (First Three) and Varying Measurement Error Standard Deviation (Last Two). The conditional density type of $T$: Uniform; The noise type of measurement error $U$: Normal.

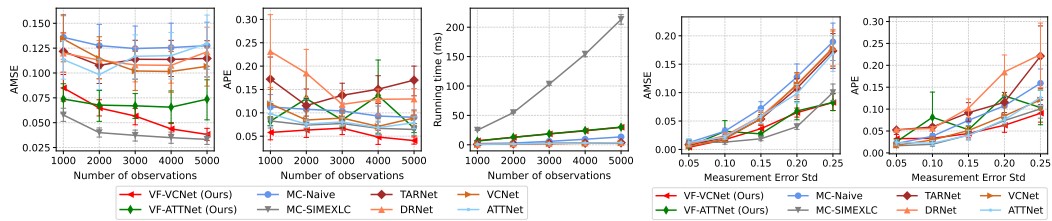

Figure 11: Comparing AMSE (↓), APE (↓), and Runtime (↓) for VF-VCNet, VF-ATTNet, and baselines on 1-D covariate simulated Data. MLP function between $T$ and $X$. Columns: Varying Observation Counts (First Three) and Varying Measurement Error Standard Deviation (Last Two). The conditional density type of $T$: Uniform; The noise type of measurement error $U$: Laplace.

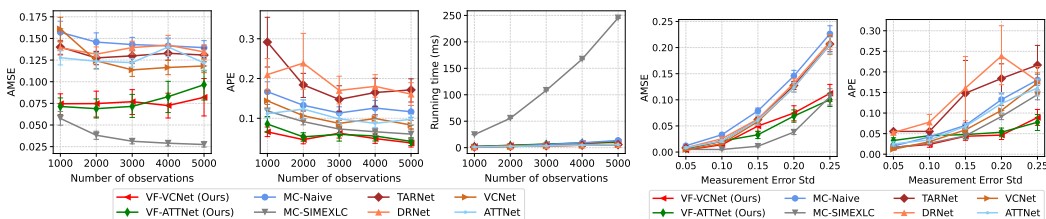

Figure 12: Comparing AMSE (↓), APE (↓), and Runtime (↓) for VF-VCNet, VF-ATTNet, and baselines on 1-D covariate simulated Data. MIM function between $T$ and $X$. Columns: Varying Observation Counts (First Three) and Varying Measurement Error Standard Deviation (Last Two). The conditional density type of $T$: Normal; The noise type of measurement error $U$: Laplace.

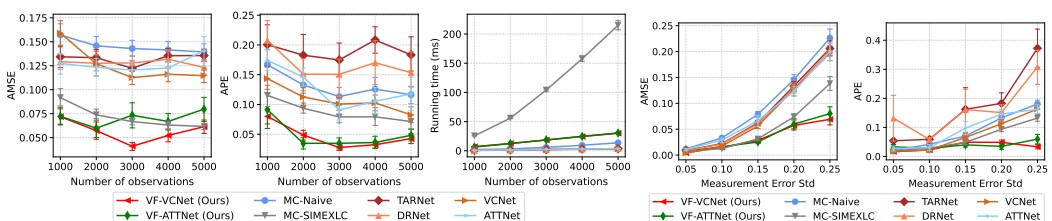

Figure 13: Comparing AMSE (↓), APE (↓), and Runtime (↓) for VF-VCNet, VF-ATTNet, and baselines on 1-D covariate simulated Data. MIM function between $T$ and $X$. Columns: Varying Observation Counts (First Three) and Varying Measurement Error Standard Deviation (Last Two). The conditional density type of $T$: Uniform; The noise type of measurement error $U$: Normal.

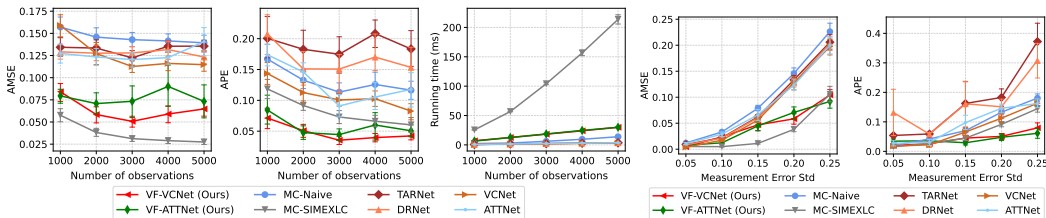

Figure 14: Comparing AMSE (↓), APE (↓), and Runtime (↓) for VF-VCNet, VF-ATTNet, and baselines on 1-D covariate simulated Data. MIM function between $T$ and $X$. Columns: Varying Observation Counts (First Three) and Varying Measurement Error Standard Deviation (Last Two). The conditional density type of $T$: Uniform; The noise type of measurement error $U$: Laplace.

### C.3 MORE COMPARISONS ON 6-D SIMULATED DATA.

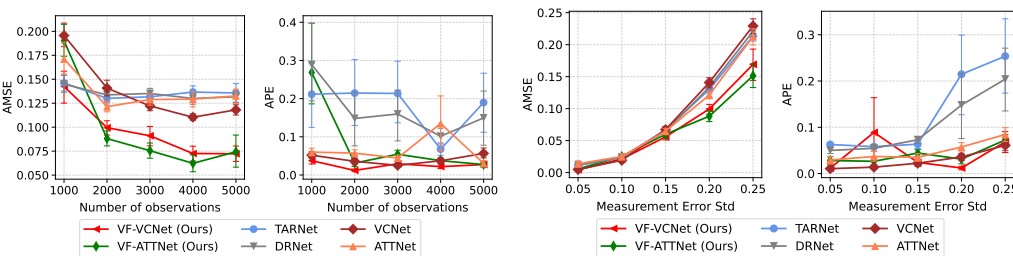

Figure 15: Comparing AMSE and APE for VF-VCNet, VF-ATTNet, and baselines on 6-D covariate Simulated Data. Row: Laplace measurement error; Columns: Varying Observation Counts (First Two) and Varying Measurement Error Standard Deviation (Last Two).

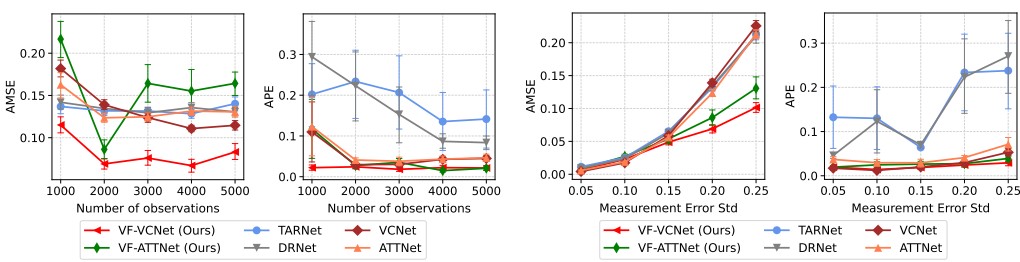

Figure 16: Comparing AMSE (↓), APE (↓), and Runtime (↓) for VF-VCNet, VF-ATTNet, and baselines on 6-D covariate simulated Data. Columns: Varying Observation Counts (First Three) and Varying Measurement Error Standard Deviation (Last Two). The conditional density type of $T$: Uniform; The noise type of measurement error $U$: Normal.

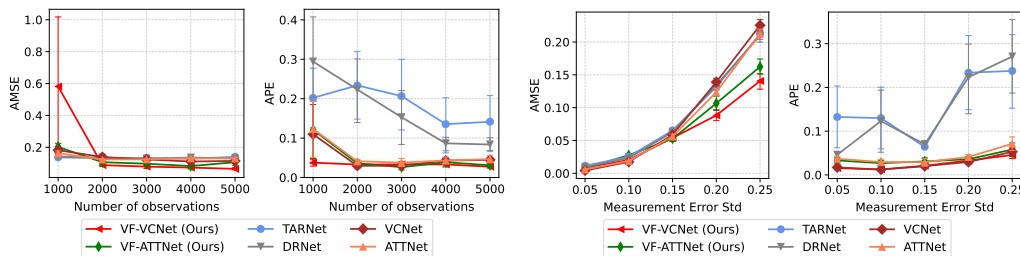

Figure 17: Comparing AMSE (↓), APE (↓), and Runtime (↓) for VF-VCNet, VF-ATTNet, and baselines on 6-D covariate simulated Data. Columns: Varying Observation Counts (First Three) and Varying Measurement Error Standard Deviation (Last Two). The conditional density type of $T$: Uniform; The noise type of measurement error $U$: Laplace.

## D    Implementation and Metrics Details

We provide the implementation details and the formulation of the two metrics used for our proposed and all baseline methods.

### D.1    Implementations

In our paper, we have adapted the conditional density estimation head for VCNet, ATTNet, and our method VF-VCNet and VF-ATTNet to accommodate the out-of-range distribution of variable $t$, which may fall outside of the interval $[0, 1]$. For all cases, we involve estimating the mean and variance parameters of a Gaussian distribution when the prior $p_{T|\mathbf{X}}(t|\mathbf{x})$ is Gaussian, or employing a Gaussian mixture distribution to better approximate non-Gaussian prior distributions. For all non-Gaussian cases, we set the number of mixture models as $4$. Additionally, in line with the recommendations in Shi et al. (2019), we have introduced a conditional density estimation head for DRNet to enhance its performance.

Following the way akin to that of (Huang & Zhang, 2023) for assessing the error of individual testing samples, we adopt a focused approach by computing the AMSE and APE within a designated interval. They consider the interval $[q_{0.1}, q_{0.9}]$ where $q_{0.1}$ and $q_{0.9}$ are the 10-th and 90-th quantiles of the treatment, respectively. To streamline our analysis for consistency and simplicity across all experiments presented in this paper, we opt to employ the interval $[-0.3, 1.3]$.

We have two experimental settings in our main paper. For the varying the number of observations case, we set the measurement error with variance $0.04$ ($0.25$ is always set for the variance of $p_{T|\mathbf{X}}(t|\mathbf{x})$.) and vary the number of observations in the training set. For the varying measurement error variance case, we set the number of observations in the training set as $2000$ and vary the measurement error variance to get the final results.

- **TARNet and DRNet**[2]: The prior estimator of TARNet and DRNet includes estimate $T$ for different intervals. To adapt to the settings in our paper, we divide the whole interval averagely into $8$ subintervals.

- **VCNet**[3]: We directly use the codes provided in their official Github Repo and keep all hyperparameters the same as used in their paper.

- **ATTNet**[4]: We directly use the codes provided in their official Github Repo and made the following modifications in our paper.

  The treatment effects estimation (TEE) method using the transformer technique, abbreviated as TransTEE, mainly relies on two parts. Firstly, it involves estimating the generalized propensity score (GPS), denoted as $p(T|\mathbf{X})$. In the TransTEE framework, there are two approaches (a direct regression version and a probabilistic version) to GPS estimation, both based on modeling $p(T|\mathbf{X})$ as a normal distribution. Unlike TransTEE, we do not impose constraints on this distribution; instead, in our framework, we treat it as a prior and employ a mixture density network for direct estimation, which can also serve as a GPS estimator. This design is applied within the proposed VF-VCNet and VF-ATTNet methods.

  The second part is predicting the outcome $Y$ conditioned on covariates $\mathbf{X}$ and treatment $T$, specifically obtaining $p(Y|T, \mathbf{X})$. In this regard, we directly take the module within TransTEE, including designing the embedding representation of $T$ and the network characterized by a stack of Attention layers, and we refer to it as ATTNet in our paper.

- **MC-Naive and MC-SIMEXLC** [5]: We directly use the codes provided in their official codes and keep all hyperparameters the same as used in their paper. However, we do not have the Python version of all codes, since we directly run the Matlab codes in a Python environment

---

[2]https://github.com/lushleaf/varying-coefficient-net-with-functional-tr
[3]https://github.com/lushleaf/varying-coefficient-net-with-functional-tr
[4]https://github.com/hlzhang109/TransTEE
[5]The codes were provided by the first author of (Huang & Zhang, 2023).

## D.2 HARDWARE

In this work, we perform all experiments on a cluster with a $64$-core Intel CPU and a total $567$ GB Memory RAM. However, we run each individual experiment with only one core.

## D.3 METRICS.

To fairly compare our proposed method and baselines, we take two well-known metrics, the Average Mean Squared Error (AMSE) (Nie et al., 2020; Zhang et al., 2022) and the Average Policy Error (APE) (Bica et al., 2020) to evaluate the performance of all methods.

The AMSE measures how well the models estimate the patient outcome across the entire dosage space:

$$\text{AMSE} = \frac{1}{N} \sum_{i=1}^{N} \int_{\mathcal{T}} \Big( y_i(\mathbf{x}, t) - \hat{y}_i(\mathbf{x}, t) \Big)^2 p(t) \mathrm{d}t \,. \tag{24}$$

In addition to this, we compute the APE (Schwab et al., 2019; Bica et al., 2020) which compares the distance of the estimated optimal treatment and the groud-truth treatment:

$$\text{APE} = \frac{1}{N} \sum_{i=1}^{N} \big| t_{\text{gt}} - t_{\text{est}} \big| \,, \tag{25}$$

where $t_{gt}$ is the ground-truth optimal treatment and $t_{est}$ is the estimated optimal treatment identified by the model. Each of these metrics is computed on a held-out test set.

# E MORE DISCUSSIONS

In this section, we provided more discussion on our method to help the audience who may have concerns on the following points.

## E.1 WHAT HAPPENS TO THE PERFORMANCE OF THE ALGORITHM IF THE TREATMENTS ARE MEASURED WITHOUT AN ERROR?

Notice that in this paper, we develop our method under the assumption that *the distribution formulation of the measurement error, denoted as $U$, is known.* When there is no measurement error present, the distribution of the measurement error becomes a discrete distribution with its probability mass concentrated solely on zero. Consequently, it allows for the direct determination of the true variable $T$ without the need for a variational step to infer $T$ from observed variables $S$ and $Y$. In this case, our variational framework would degenerate to the basic model that directly takes $T$ to estimate the treatment effect.

In real-world scenarios (Carroll et al., 2006), researchers usually devise specific methods to deal with measurement error in situations where there is a confirmed presence of measurement error in the data. This suspicion typically arises when there are inconsistent multiple measurements of a variable for a single individual. Furthermore, the existence of multiple measurements enables the identification of characteristic functions associated with the measurement error. Building upon this fact, our proposed method capitalizes on prior knowledge of the measurement error and strategically maximizes the joint likelihood to get an unbiased estimator for treatment effects.

## E.2 THE DIFFERENCE BETWEEN OUR METHOD AND THE EXISTING VAE METHODS FOR TREATMENT EFFECT ESTIMATION.

In the realm of treatment effect estimation, a fundamental challenge is addressing the selection bias, rooted in confounding variables that influence both treatment choice and outcomes. Machine learning approaches typically branch into two methods: discriminative methods (Nie et al., 2020; Zhang et al., 2022) rely on Generalized Propensity Score (GPS) to extract the confounders, while generative methods (Hassanpour & Greiner, 2019) design a latent generative model for covariates and aim to identify the latent variables as you noted.

For generative models, they assume that the covariates are generated by latent confounding variables and adjustment variables via a mixing function. The adjustment variables only affect the outcome values. Therefore, the key of this method is to (1) recover/identify the latent factors, (2) and separate the confounding variables, which inevitably raise the model identifiability issue. As you rightly pointed out, Nonlinear Independent Component Analysis (ICA) (Khemakhem et al., 2020) techniques achieve model identifiability by introducing distribution assumptions on latent variables relying on some auxiliary information. Moreover, recent research (Von Kügelgen et al., 2021) also explores theoretical guarantees for separating latent factors, offering promising contributions to this challenge.

In our approach, we primarily employ the GPS to extract the confounding variables in the treatments, aligning with the discriminative methods. However, our differentiator lies in utilizing the variational framework, focusing on the unobservability of the treatment variable rather than covariate-related issues. We leverage variational methods because our learning objective—specifically, the marginal log-likelihood of observed variables, entails an integral over the unobserved treatment, presenting a computational challenge for analytical solutions.

## E.3 ADDITIONAL DISCUSSIONS ON THE EXPERIMENTAL RESULTS IN THE MAIN PAPER.

*Why do non-VI methods perform well when we have a low measurement error?*

The presence of measurement error (ME) can introduce bias in estimating the true underlying parameters, and this bias tends to decrease as the measurement error diminishes (Carroll et al., 2006). Therefore, non-VI methods would be expected to perform well when we have a low ME. Notably, some previous works (Hu et al., 2022; Huang & Zhang, 2023) also directly take the methods which ignore the measurement error as baseline methods.

*In our experimental results, why do the non-VI approaches perform basically on par?*

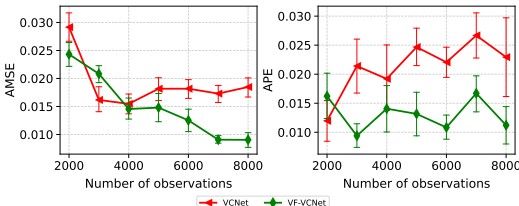

Figure 18: Comparing AMSE (↓), APE (↓) for VCNet and VF-VCNet, and baselines on 6-D covariate simulated Data. MLP function between $T$ and $X$. The standard deviation of the measurement error is set to $0.1$.

In our experiment involving "varying ME standard deviation", we systematically increased the standard deviation of the measurement error to illustrate the consistent advantages of our method. As you correctly noted, our approach exhibits performance gains, particularly when the ME is relatively high ($\geq 2.0$). For lower errors ($0.5, 1.0, 1.5$), the non-VI methods perform on par with that of our methods. There are two primary reasons for this case. (1) The impact of measurement error on non-VI methods is expected to be small when the ME is low. (2) While our method is expected to outperform non-VI methods since it provides an unbiased estimator for treatment effect, it is essential to note that our method introduces additional parameters to infer the treatment from measured variables. This complexity in the learning process necessitates more data to fit the whole model.

Therefore, when the number of observations is limited ($2000$ for this setting in our paper), the performance of non-VI methods and our methods may be comparable.

*With more data, our method can perform better with a low measurement error.*

As we discussed, our method provides an unbiased estimator for the treatment effect, albeit requiring more data. To verify this, we added an experiment when the ME was relatively low (std=1.0), where we systematically varied the number of observations to compare our method with a non-VI method. For the sake of illustration, we present the results (Average MSE) of VCNet and VF-VCNet (ours) for example. The results are shown in Fig. (18).

Based on the above results, it is evident that VCNet tends to converge to a biased model as the number of observations increases. In contrast, the AMSE of our method consistently decreases with a larger number of observations.

*Results on semi-synthetic data show our proposed method is ineffective.*

According to these results on all three synthetic datasets, our methods outperform the baseline methods on the IHDP and TCGA datasets. (Notice that our two methods should be compared with their respective base methods, namely VF-VCNet versus VCNet and VF-ATTNet versus ATTNet.)

On the News dataset, we observed that VF-VCNet continues to exhibit a slight advantage over VCNet. However, VF-ATTNet performs worse than its basleine method ATTNet, for two key reasons. (1) The parameters of the VF-ATTNet model are larger compared to other models, necessitating more data for effective learning. However, the News dataset comprises only 3000 observations with $495$ covariates. (2) The simulated conditional distribution $p(T|\mathbf{X})$ is a complex beta distribution (compared to IHDP and TCGA), with both sufficient statistics being flexible and dependent on $\mathbf{X}$. For our methods, since we lack the treatment $T$ to supervise the learning of $p(T|\mathbf{X})$, we treat this term as a prior to learning which poses a challenge. Therefore, our VF-ATTNet underperforms its base model, ATTNet.

*Why do we choose to use Neural network-based methods?*

We assert that employing NN offers two distinct advantages over nonparametric methods, particularly in addressing high-dimensional issues and enhancing computing efficiency. This claim is supported by our simulation results, where our method consistently demonstrates performance gains compared to MC-SIMEXLC, and notably, the running time of our approach is significantly lower.

### E.4 MORE RESULTS TO SHOW THE EFFECTIVENESS OF OUR PROPOSED METHODS IN THE HIGH-DIMENTIONAL SETTINGS.

We can see that in the experimental results shown in Table 1. Our proposed method VF-ATTNet fails to surpass the performance of its base method ATTNet. ATTNet leverages attention mechanisms to learn embeddings of covariates and treatment, as well as capture their interactions during outcome prediction. It's worth noting that integrating attention mechanisms typically demands more observations for effective model learning.

In the case of IHDP, with only $747$ observations available, and a subset of $598$ used for training, we acknowledge that the sample size falls short of meeting the necessary conditions for attention-based methods to work well.

Another factor affecting the model performance is the complexity of the conditional distribution, which poses a challenge to identify the true treatment effect in the presence of measurement error. In Table 1, concerning the News dataset, both our proposed methods VF-VCNet, and VF-ATTNet, and their base methods VCNet, and ATTNet exhibit higher AMSE when applied to this dataset, underscoring the difficulty posed by the dataset's characteristics. These results effectively highlight the challenges in estimating treatment effects with measurement error, also the limitations of our proposed methods.

Therefore, we further elaborated three more experiments, including two high-dimensional synthetic datasets and a modified IHDP dataset, to show the advantage of our proposed methods over their base methods and baseline methods.

In generating synthetic data, we adopt the same methodology used in our previous simulations to create covariates with dimensions of 50 and 100. The treatment is generated given the covariates through a fixed MLP function. Then, we rescale the output to an interval by $t = (1 + \exp(-\tilde{t}))^{-1}$ and add a random noise sampled from a Gaussian distribution $\mathcal{N}(0, 0.25)$ to obtain the treatment. Then, we add the measurement error to get the error-contaminated measurement $S$. To get the predicted outcome, we adopted the same way as the first dose-response function used in (Bica et al., 2020). We first generate a set of parameters $\mathbf{v}_1, \mathbf{v}_2, \mathbf{v}_3$. For each run of the experiment, these parameters are sampled randomly by sampling a vector, $\mathbf{u}$, from $\mathcal{N}(\mathbf{0}, \mathbf{1})$ and then setting $\mathbf{v} = \mathbf{u}/\|\mathbf{u}\|$ where $\|\cdot\|$ is Euclidean norm.

$$f(\mathbf{x}, s) = C((\mathbf{v}_1)^T \mathbf{x} + 12(\mathbf{v}_2)^T \mathbf{x} t - 12(\mathbf{v}_3)^T \mathbf{x} t^2) \tag{26}$$

where $t$ is the corresponding treatment value and $\mathbf{x}$ represents the covariates vector. Following this data generation process, we create a dataset comprising $6000$ observations, with $4800$ designated for training. For the IHDP dataset, we followed the same data generation procedures outlined in (Nie et al., 2020; Zhang et al., 2022), including the implementation $p(T|\mathbf{X})$ and $p(Y|T, \mathbf{X})$. Covariates $X$ of 25 dimensions followed the original size in the IHDP dataset were simulated using the method employed in our simulations, involving sampling from a Uniform distribution over the interval $[0, 1]$. Subsequently, following this procedure, we generated a total of $4000$ observations.

Table 2: Performance of ADRF on three high-dimensional datasets. Metrics are reported as Mean ± Std.

| Method | Simulation (d=50) | | Simulation (d=100) | | Modified IHDP | |
| --- | --- | --- | --- | --- | --- | --- |
| | AMSE | APE | AMSE | APE | AMSE | APE |
| TARNet | $0.066 \pm 0.020$ | $0.130 \pm 0.072$ | $0.114 \pm 0.031$ | $0.145 \pm 0.042$ | $0.125 \pm 0.032$ | $0.076 \pm 0.031$ |
| DRNet | $0.067 \pm 0.017$ | $0.142 \pm 0.041$ | $0.121 \pm 0.040$ | $0.149 \pm 0.047$ | $0.112 \pm 0.015$ | $0.085 \pm 0.050$ |
| ATTNet | $0.059 \pm 0.021$ | $0.53 \pm 0.015$ | $0.129 \pm 0.036$ | $0.063 \pm 0.031$ | $0.138 \pm 0.032$ | $0.101 \pm 0.041$ |
| VCNet | $0.068 \pm 0.028$ | $0.052 \pm 0.030$ | $0.103 \pm 0.024$ | $0.093 \pm 0.044$ | $0.110 \pm 0.015$ | $0.108 \pm 0.045$ |
| VF-ATTNet | $0.032 \pm 0.010$ | $0.034 \pm 0.022$ | $0.085 \pm 0.018$ | $0.041 \pm 0.025$ | $0.066 \pm 0.019$ | $0.061 \pm 0.038$ |
| VF-VCNet | $0.044 \pm 0.022$ | $0.028 \pm 0.025$ | $0.087 \pm 0.026$ | $0.072 \pm 0.034$ | $0.059 \pm 0.017$ | $0.047 \pm 0.017$ |

According to these results, our proposed methods, both VF-VCNet and VF-ATTNet, outperform their base models and all the other baseline methods on the high-dimensional datasets.

