# OpenReview forum: "A Variational Framework for Estimating Continuous Treatment Effects with Measurement Error"
_ICLR.cc/2024/Conference — ICLR 2024 poster_

### Official Review · Reviewer_2KQn · 2023-10-31

**Soundness:** 3 good
**Presentation:** 3 good
**Contribution:** 3 good
**Rating:** 8
**Confidence:** 4

**Summary:**

The paper proposes a variational framework to estimate the causal effect of a continuous treatment in the presence of measurement error, with the unobserved true treatment as a latent variable to be learnt. Experiments on synthetic data demonstrate decent performance in some settings.

**Strengths:**

- Interesting setting not often considered in TEE literature (at least my knowledge of recent literature).
- VI approach seems to make sense and pretty standard.
- Well written, meme is nice as an illustration of the problem.

**Weaknesses:**

- It's not really apparently where the measurement errors comes into play (more concretely I'm not sure the DAG makes sense to me). To take your meme example: Doctor prescribes S, which was prescribed due to X, and the resulting treatment as a result of the patient not adhering to the prescription is T. The resulting outcome is Y, with measurement error too. So wouldn't the DAG be: X->S | S+U->T | X + T + eps -> Y
- Is ATTNet different from TransTEE, because the actual name from the paper.
- The experimental results section is disappointing. On synthetic datasets, unless you have a high measurement (which I'm not sure if completely reasonable but regardless), the non-VI approaches perform basically on par. But semi-synthetic really shows the this method is ineffective: VI underperforms significantly. And only reason to use NN is when there is when there is high-dimensions, so not sure what the point of this method is then.

Minor Comments:
- Plots are hard to read when printed. Perhaps less graphs, but bigger?

**Questions:**

Just a response to the above.

---

> ### Author Response · Authors · 2023-11-18
> **Response to reviewer 2KQn (part1/3)**
>
> We sincerely appreciate your insightful comments, which significantly enhance our paper. Your feedback enables us to provide a clearer elucidation of our method's advantages and limitations.
>
> **[Q1. It's not really apparently where the measurement errors comes into play (more concretely I'm not sure the DAG makes sense to me). To take your meme example: Doctor prescribes S, which was prescribed due to X, and the resulting treatment as a result of the patient not adhering to the prescription is T. The resulting outcome is Y, with measurement error too. So wouldn't the DAG be: X->S | S+U->T | X + T + eps -> Y]**
>
> Thanks for this valuable suggestion for the improper meme we introduced in Fig.1. According to the current meme, the corresponding DAG would be as you suggested instead of our focused one.
>
> In our paper, we focus on continuous treatment data measured with classical error; instead of observing the received true treatment, researchers only observe the sum of the treatment and a random error, which mainly comes from the error reporting issue. The real case of this problem can be found in the Epidemiologic Study Cohort data from the first National Health and Nutrition Examination (NHANES-I) [1,2], where over $75\%$ of the variance in the fat intake data is made up of measurement error since the daily saturated fat intake was measured using a single 24-hour recall. In our paper, we focus on continuous treatment data measured with classical error; instead of observing the received treatment, researchers only observe the sum of the treatment and a random error.
>
> To correct this mistake, we have updated the example and provided a new meme in the revision to avoid misunderstanding our objective.
>
> **[Q2. Is ATTNet different from TransTEE, because the actual name from the paper.]**
>
> The treatment effects estimation (TEE) method using the transformer technique, abbreviated as TransTEE, mainly relies on two parts. Firstly, it involves estimating the generalized propensity score (GPS), denoted as $p(T|\mathbf{X})$. In the TransTEE framework, there are two approaches (a direct regression version and a probabilistic version) to GPS estimation, both based on modeling $p(T|\mathbf{X})$ as a normal distribution. Unlike TransTEE, we do not impose constraints on this distribution; instead, in our framework, we treat it as a prior and employ a mixture density network for direct estimation, which can also serve as a GPS estimator. This design is applied within the proposed VF-VCNet and VF-ATTNet methods.
>
> The second part is predicting the outcome $Y$ conditioned on covariates $\mathbf{X}$ and treatment $T$, specifically obtaining $p(Y|T,\mathbf{X})$. In this regard, we directly take the module within TransTEE, including designing the embedding representation of $T$ and the network characterized by a stack of Attention layers, and we refer to it as ATTNet in our paper.
>
> In our revision, we will provide additional clarity on these methodologies to enhance the understanding of our framework.

---

> > ### Comment · Reviewer_2KQn · 2023-11-21
> > **Q1+2**
> >
> > Q1. Looking at the DAG again this makes sense. Thanks!
> > Q2. Noted, thanks!

---

> ### Author Response · Authors · 2023-11-18
> **Response to reviewer 2KQn (part2/3)**
>
> **[Q3. The experimental results section is disappointing. On synthetic datasets, unless you have a high measurement (which I'm not sure if completely reasonable but regardless), the non-VI approaches perform basically on par. But semi-synthetic really shows the this method is ineffective: VI underperforms significantly. And only reason to use NN is when there is when there is high-dimensions, so not sure what the point of this method is then.]**
>
> We politely disagree that our experimental results are disappointing, as we believe the perceived disappointment may stem from an insufficient clarification of the limitations of our methods. After reading your comment, we think it should be the unclearificaiton of the limitations of our methods lead to all these confusion. We intend to address your concerns point by point.
>
> > Unless you have a high measurement (which I'm not sure if completely reasonable but regardless), the non-VI approaches perform basically on par.
>
> We partially agree with this argument that our method is only useful in the presense of a high measurement error. We contend that its advantages extend even when the measurement error is relatively low. We clarify this argument from three points as follows.
>
> (1) *Why do non-VI methods perform well when we have a low measurement error?*
>
> The presence of measurement error (ME) can introduce bias in estimating the true underlying parameters, and this bias tends to decrease as the measurement error diminishes [1]. Therefore, non-VI methods would be expected to perform well when we have a low ME. Notably, some previous works [2,3] also directly take the methods which ignore the measurement error as baseline methods.
>
> (2) *In our experiemental results, why does the non-VI approaches perform basically on par.*
>
> In our experiment involving "varying ME standard deviation", we systematically increased the standard deviation of the measurement error to illustrate the consistent advantages of our method. As you correctly noted, our approach exhibits performance gains, particularly when the ME is relatively high (std$\geq0.2$). For lower errors ($0.05, 0.1, 0.15$), the non-VI methods perform on par with that of our methods. There are two primary reasons for this case.
>
> - The impact of measurement error on non-VI methods is expected to be small when the ME is low.
> - While our method is expected to outperform non-VI methods since it provides an unbiased estimator for treatment effect, it is essential to note that our method introduces additional parameters (see Figure 6d) to infer the treatment from measured variables. This complexity in the learning process necessitates more data to fit the whole model.
>
> Therefore, when the number of observations is limited ($2000$ for this setting in our paper), the performance of non-VI methods and our methods may be comparable.
>
> (3) *With more data, our method can perform better with a low measurement error.*
>
> As we discussed, our method provides an unbiased estimator for the treatment effect, albeit requiring more data. To verify this, we added an experiment when the ME is realtively low (std=0.1), where we systematically varied the number of observations to compare our method with a non-VI method. For the sake of illustration, we present the results (Average MSE) of VCNet and VF-VCNet (ours) for example. The results are detailed below:
>
> | Number of observatioins| VCNet (AMSE) |VF-VCNet (AMSE) |
> |------|--------------------|-------------------|
> | 2000 |   0.029 ± 0.009    | 0.024 ± 0.007     |
> | 3000 |   0.016 ± 0.007    | 0.021 ± 0.005     |
> | 4000 |   0.015 ± 0.006    | 0.015 ± 0.006     |
> | 5000 |   0.018 ± 0.007    | 0.015 ± 0.009     |
> | 6000 |   0.018 ± 0.006    | 0.013 ± 0.007     |
> | 7000 |   0.017 ± 0.005    | 0.009 ± 0.002     |
> | 8000 |   0.018 ± 0.006    | 0.008 ± 0.004     |
>
> Based on the above results, it is evident that VCNet tends to converge to a biased model as the number of observations increases. In contrast, the AMSE of our method consistently decreases with a larger number of observations.

---

> ### Author Response · Authors · 2023-11-18
> **Response to reviewer 2KQn (part3/3)**
>
> **[Q3. Continuing]**
>
> > But semi-synthetic really shows the this method is ineffective: VI underperforms significantly.
>
> In our paper, we have included two semi-synthetic datasets, namely IHDP and News. Responding to the request by **Reviewer gZZT**, we have also added a large dataset named TGCA, including gene information from $9659$ patients, in the revision. The results of this new dataset are detailed below:
>
> | Method          | TCGA (AMSE)      | TCGA (APE)      |
> |-----------------|------------------|-----------------|
> | TARNet          | 0.042 ± 0.007    | 0.136 ± 0.069   |
> | DRNet           | 0.047 ± 0.007    | 0.182 ± 0.040   |
> | ATTNet          | 0.056 ± 0.019    | 0.076 ± 0.050   |
> | VCNet           | 0.046 ± 0.010    | 0.067 ± 0.045   |
> | VF-ATTNet(Ours) | 0.042 ± 0.010    | 0.057 ± 0.037   |
> | VF-VCNet (Ours) |**0.019 ± 0.014** |**0.036 ± 0.017**|
>
> According to these results, our methods outperform the baseline methods on the IHDP and TCGA datasets. (Notice that our two methods should be compared with their respective base methods, namely VF-VCNet versus VCNet and VF-ATTNet versus ATTNet.)
>
> On the News dataset, we observed that VF-VCNet continues to exhibit a slight advantage over VCNet. However, VF-ATTNet performs worse than its basleine method ATTNet, for two key reasons. (1) The parameters of the VF-ATTNet model are larger compared to other models, necessitating more data for effective learning. However, the News dataset comprises only $3000$ observations with $495$ covariates. (2) The simulated conditional distribution $p(T|\mathbf{X})$ is a complex beta distribution (compared to IHDP and TCGA), with both sufficient statistics being flexible and dependent on $\mathbf{X}$. For our methods, since we lack the treatment $T$ to supervise the learning of $p(T|\mathbf{X})$, we treat this term as a prior to learning which poses a challenge. Therefore, our VF-ATTNet underperforms its base model, ATTNet.
>
> > And only reason to use NN is when there is when there is high-dimensions, so not sure what the point of this method is then.
>
> We assert that employing NN offers two distinct advantages over nonparametric methods, particularly in addressing **high-dimensional issues** and **enhancing computing efficiency**. This claim is supported by Figure 3, where our method consistently demonstrates performance gains compared to MC-SIMEXLC, and notably, the running time of our approach is significantly lower.
>
> We believe your confusion may have arisen due to our missing discussion on the limitations of our methods. In response to this, we have revised our paper to include a more comprehensive examination of the limitations, aiming to provide clarity on this aspect.
>
>
> **[Q4. Figure issues.]**
>
> Thank you for your suggestion. We have resized the figures and restructured the layout to enhance readability. Additionally, certain figures have been relocated to the Appendix for improved clarity.
>
> [1] [Carroll R. J., Rupper D., Stefanski L. A., Crainiceanu C. M. (2006). Measurement error in nonlinear models: A modern perspective. Chapman Hall/CRC.](https://www.taylorfrancis.com/books/mono/10.1201/9781420010138/measurement-error-nonlinear-models-ciprian-crainiceanu-raymond-carroll-leonard-stefanski-david-ruppert)
>
> [2] [Huang, W. and Zhang, Z. (2023). Nonparametric estimation of continuous treatment effect with measurement error. JRSSB, 85, 474 - 496.](https://academic.oup.com/jrsssb/article/85/2/474/7098034)
>
> [3] [Hu, Zhirui, Zheng Tracy Ke, and Jun S. Liu. Measurement error models: from nonparametric methods to deep neural networks. Statistical Science 37, no. 4 (2022): 473-493.](https://projecteuclid.org/journals/statistical-science/volume-37/issue-4/Measurement-Error-Models--From-Nonparametric-Methods-to-Deep-Neural/10.1214/21-STS834.short)

---

> ### Comment · Reviewer_2KQn · 2023-11-21
> **Q3**
>
> (1) Why do non-VI methods perform well when we have a low measurement error?
> (2) In our experiemental results, why does the non-VI approaches perform basically on par.
> (3) With more data, our method can perform better with a low measurement error.
>
> Sure, I don't think I disagree with anything you said. I think a contextualisation of the results is helpful. However, I think my point stands: there are no results showing gains in a high-dimensional setting. I don't know if I misread the results, but I see that ATTNet outperforms VF-ATTNet on IHDP and News. The TCGA results look good, but ATTNet is within the standard deviation of VF-ATTNet. What I would like to see is good performance on a high-dimensional synthetic or semi-synthetic dataset.

---

> > ### Author Response · Authors · 2023-11-22
> > **Reponse to Reviewer 2KQn (part 1/2)**
> >
> > We sincerely apologize for our misunderstanding of your previous comments, specifically regarding the request for additional results showcasing how our proposed method can outperform the baseline methods. Thank you for providing the additional comments. We believe incorporating your suggestions will significantly enhance the overall quality of our paper.
> >
> > We are actively working to address the concerns you raised, regarding (1) "there are no results showing gains in a high-dimensional setting". (2) I don't know if I misread the results, but I see that ATTNet outperforms VF-ATTNet on IHDP and News. (3) What I would like to see is good performance on a high-dimensional synthetic or semi-synthetic dataset. point by point below.
> >
> > > There are no results showing gains in a high-dimensional setting.
> >
> > To address the challenge of treatment effects estimation in the presence of measurement error, we introduce a variational framework capable of integrating various error-ignoring methods as base models. Specifically, this paper incorporates two base methods: VCNet and ATTNet. Note that both VF-VCNet and VF-ATTNet are considered as our proposed methods within this framework.
> >
> > As you can see, in all simulation scenarios, VF-VCNet consistently exhibits substantial performance improvements over its base model, VCNet, and the other baseline methods. Notably, it achieves the best results across both the IHDP and TCGA datasets, both characterized by high-dimensional settings. This underscores the robust performance of VCNet as a base model when confronted with high-dimensional data, showcasing both sample efficiency and effectiveness. These results also help to verify that our proposed framework demonstrates its efficacy (even in high-dimensional settings) by incorporating VCNet as the base method.
> >
> > Moreover, according to your kind suggestion, we have included two additional datasets with high-dimensional covariates below to further illustrate the effectiveness of our proposed methods.
> >
> > > I don't know if I misread the results, but I see that ATTNet outperforms VF-ATTNet on IHDP and News.
> >
> > Yes, the information you've encountered is correct. In our current results on the IHDP and News datasets, ATTNet outperforms VF-ATTNet. We believe that this may be the source of your concerns. As explained earlier, ATTNet leverages attention mechanisms to learn embeddings of covariates and treatment, as well as capture their interactions during outcome prediction. It's worth noting that integrating attention mechanisms typically demands more observations for effective model learning.
> >
> > In the case of IHDP, with only 747 observations available, and a subset of 598 used for training, we acknowledge that the sample size falls short of meeting the necessary conditions for attention-based methods to work well.
> >
> > Another factor affecting the model performance is the complexity of the conditional distribution $p(T|\mathbf{X})$, which poses a challenge to identify the true treatment effect in the presence of measurement error. In Table 1, concerning the News dataset, both our proposed methods VF-VCNet, VF-ATTNet, and their base methods VCNet, ATTNet exhibit higher AMSE when applied to this dataset, underscoring the difficulty posed by the dataset's characteristics.
> >
> > These results effectively highlight the challenges in estimating treatment effects with measurement error, also the limitations of our proposed methods.

---

> ### Author Response · Authors · 2023-11-22
> **Reponse to Reviewer 2KQn (part 2/2)**
>
> **Continuing...**
> > What I would like to see is good performance on a high-dimensional synthetic or semi-synthetic dataset. point by point below. (can our porposed method work well in the high-dimensional settings?)
>
> We further elaborated three more experiments, including two synthetic high-dimensional datasets and a modified IHDP dataset, to show the advantage of our proposed methods over their base methods and baseline methods.
>
> In generating synthetic data, we adopt the same methodology used in our previous simulations to create covariates with dimensions of 50 and 100. The treatment is generated given the covariates through a fixed MLP function. To get the predicted outcome, we adopted the same way as the first dose-response function used in [3]. Following this data generation process, we create two datasets ($d _{\mathbf{X}}=50$ and $d _{\mathbf{X}}=100$) comprising 6000 observations, with 4800 used for training. We then evaluate all methods on this dataset, and the results are presented below:
>
> | Method          | Simulation d=50 (AMSE)| Simulation d=50 (APE)| Simulation d=100 (AMSE)| Simulation d=100 (APE)|
> |-----------------|------------------|-----------------|------------------|-----------------|
> | TARNet          | 0.066 ± 0.020    | 0.130 ± 0.072   |0.114 ± 0.031    | 0.145 ± 0.042   |
> | DRNet           | 0.067 ± 0.017    | 0.142 ± 0.041   |0.121 ± 0.040    | 0.149 ± 0.047   |
> | ATTNet          | 0.059 ± 0.021    | 0.053 ± 0.015   |0.129 ± 0.036    | 0.063 ± 0.031   |
> | VCNet           | 0.068 ± 0.028    | 0.052 ± 0.030   |0.103 ± 0.024    | 0.093 ± 0.044   |
> | VF-VCNet (Ours) | 0.044 ± 0.022    | 0.028 ± 0.025   |0.087 ± 0.026    | 0.072 ± 0.034   |
> | VF-ATTNet(Ours) | 0.032 ± 0.010    | 0.034 ± 0.022   |0.085 ± 0.018    | 0.041 ± 0.025   |
>
>
> Then, we followed the same data generation procedures for the IHDP dataset outlined in [1,2], including the implementation $p(T|\mathbf{X})$ and $p(Y|T, \mathbf{X})$. Covariates $X$ of 25 dimensions followed the original size in the IHDP dataset were simulated using the method employed in our simulations, involving sampling from a Uniform distribution over the interval [0,1]. Subsequently, following this procedure, we generated a total of 4000 observations. All methods were applied to this dataset, and the detailed results on this modified dataset are presented below:
>
> | Method          | modified IHDP (AMSE)| modified IHDP (APE)|
> |-----------------|------------------|-----------------|
> | TARNet          | 0.125 ± 0.032    | 0.076 ± 0.031   |
> | DRNet           | 0.112 ± 0.015    | 0.085 ± 0.050   |
> | ATTNet          | 0.138 ± 0.032    | 0.101 ± 0.041   |
> | VCNet           | 0.110 ± 0.015    | 0.108 ± 0.045   |
> | VF-ATTNet(Ours) | 0.066 ± 0.019    | 0.061 ± 0.038   |
> | VF-VCNet (Ours) | 0.059 ± 0.017    | 0.047 ± 0.017   |
>
> According to these results, our proposed methods, both VF-VCNet and VF-ATTNet, outperform their base models and all the other baseline methods on the high-dimensional datasets. We have added these results and the discussions in Appendix E.4 to the revision.
>
>
> [1] [Lizhen Nie, Mao Ye, Dan Nicolae, et al. Vcnet and functional targeted regularization for learning causal effects of continuous treatments. In International Conference on Learning Representations, 2020.](https://openreview.net/pdf?id=RmB-88r9dL)
>
> [2] [YiFan Zhang, Hanlin Zhang, Zachary Chase Lipton, Li Erran Li, and Eric Xing. Exploring transformer backbones for heterogeneous treatment effect estimation. In NeurIPS ML Safety Workshop, 2022.](https://arxiv.org/abs/2202.01336)
>
> [3] [Bica, Ioana, James Jordon, and Mihaela van der Schaar. Estimating the effects of continuous-valued interventions using generative adversarial networks. Advances in Neural Information Processing Systems 33 (2020): 16434-16445.](https://proceedings.neurips.cc/paper/2020/file/bea5955b308361a1b07bc55042e25e54-Paper.pdf)

---

> > ### Comment · Reviewer_2KQn · 2023-11-22
> > **Appreciate the effort!**
> >
> > Thanks for the positive response! Sincerely appreciate the extensive response to a rather short comment.
> >
> > The additional results are convincing and I think lend additional credence the performance of method in the intended application. Hope they're incorporated in the paper, as I've often seen papers in this field really not showing the efficacy of these methods on high-dimensional datasets.
> >
> > The paper now feels well rounded: tackles a novel and important setting (TEE with measurement error), novel method with experiments that show it's performance in the intended application area.
> >
> > I've updated my score to reflect this. Good work.

---

### Official Review · Reviewer_bqN4 · 2023-11-01

**Soundness:** 3 good
**Presentation:** 3 good
**Contribution:** 3 good
**Rating:** 8
**Confidence:** 4

**Summary:**

This paper proposes a variational framework for estimating the average dose-response function (ADRF) in the presence of measurement error in the treatment variable. To do this, it formulates a probabilistic model treating the unobserved true treatment as a latent variable. This allows measurement error to be handled in a principled Bayesian way. To approximate the posterior, it uses a variational inference approach to avoid expensive MCMC methods. The method can leverage state-of-the-art neural networks for ADRF estimation in the error-free setting by integrating them into the variational framework. Extensive simulation studies and experiments on semi-synthetic datasets demonstrate the efficacy of the proposed framework compared to recent nonparametric methods.

**Strengths:**

## originality
- The paper tackles the important problem of treatment effect estimation with measurement error, which has received comparatively limited attention.
- The proposed method seems novel.

## quality
- The work is well motivated and the methodology technically sound.
- Experiments are demonstrating the efficacy of the proposed methods sufficiently.

## clarity
- The paper is well written and easy to follow.

## significance
- Estimating treatment effects from observational data has many crucial real-world applications. Being able to handle measurement error robustly significantly expands the applicability of these methods. The flexible modeling framework could pave the way for practical use in areas like medicine, social sciences, and policy evaluation.

**Weaknesses:**

While overall quite strong, there are a few areas where the paper could potentially be improved:
- VAE suffers from non-identifiability. Although beta VAE might help disentangling the factors of variations, it doesn’t provide a guarantee. This is a potential limitation that is not discussed.
- Plots have very small fonts and proposed method is not marked (e.g. “ours”) making it difficult to read.

**Questions:**

(apologies for repeating some points from weaknesses)
- Can you please comment / discuss on the non-identifiability of VAEs? Could identifiability improved by the use of some auxiliary information, similarly to https://proceedings.mlr.press/v108/khemakhem20a/khemakhem20a.pdf?
- Plots are very small, please update the fonts to make it more readable. Also mark your method as "ours" to make sure the reader sees what they compare.
- nitpick: Data generation model is usually called data generating process. Not sure if you should change to this, but worth mentioning it.
- Figure 2. it's common to also add the parameters of the models in the plot (\theta_y, \theta_t, \phi). e.g. check also the original VAE paper.
- Figure 3. What does MiM stands for?
- Metrics: "we employ the widely recognized metric" - can you please provide citation?
- "this metric quantifies the proximity of our predicted outcomes to the true potential outcomes when considering covariates and the corresponding treatment" -> from the definition of the metric this doesn't quantify the error for the corresponding treatment but for all possible treatments in T. Please update the text.
- Figure 4. True response seems same between all plots, is this correct? So in that sense it's one simulation dataset but for different noise distributions?
- Table 1. Why APE is 0.0 +- 0.0? The model seem to have high AMSE error but no error (!) on APE. Would be curious to understand what happened there.
- Appendix B.4 last sentence. Fix the citation (?)

---

> ### Author Response · Authors · 2023-11-18
> **Response to reviewer bqN4 (part 1/2)**
>
> Thank you so much for your efforts in evaluating our manuscript. In the following, we will respond to your concerns point by point. Some of the responses have been updated in our revision to improve the quality of our paper.
>
> **[Q1. VAE suffers from non-identifiability. Although beta VAE might help disentangling the factors of variations, it doesn’t provide a guarantee. This is a potential limitation that is not discussed. Can you please comment / discuss on the non-identifiability of VAEs? Could identifiability improved by the use of some auxiliary information, similarly to iVAE.]**
>
> Thank you for your insightful suggestion, prompting us to refine the difference between our variational framework and the existing treatment effect estimation (TEE) methods based on Variational Autoencoders (VAEs).
>
> In the realm of TEE, a fundamental challenge is addressing the selection bias, rooted in confounding variables that influence both treatment choice and outcomes. Machine learning approaches typically branch into two methods: discriminative methods [2] rely on Generalized Propensity Score (GPS) to extract the confounders, while generative methods [1] design a latent generative model for covariates and aim to identify the latent variables as you noted.
>
> In our approach, we primarily employ GPS to extract the confounder, aligning with the discriminative methods. However, our differentiator lies in utilizing the variational framework, focusing on the unobservability of the treatment variable rather than covariate-related issues. We leverage variational methods because our learning objective—specifically, the marginal log-likelihood of observed variables, entails an integral over the unobserved treatment, presenting a computational challenge for analytical solutions.
>
> For generative models, they assume that the covariates are generated by latent confounding variables and adjustment variables via a mixing function. The adjustment variables only affect the outcome values. Therefore, the key of this method is to (1) recover/identify the latent factors, (2) and separate the confounding variables, which inevitably raise the model identifiability issue. As you rightly pointed out, Nonlinear Independent Component Analysis (ICA) [3] techniques achieve model identifiability by introducing distribution assumptions on latent variables relying on some auxiliary information. Moreover, recent research [4] also explores theoretical guarantees for separating latent factors, offering promising contributions to this challenge.
>
> We have incorporated these refinements into the Appendix of the revision to help the audience who may have similar concerns on this point. Thank you for guiding us towards a more comprehensive explanation.

---

> ### Author Response · Authors · 2023-11-18
> **Response to reviewer bqN4 (part 2/2)**
>
> **[Q2. Plots have very small fonts and proposed method is not marked (e.g. "ours") making it difficult to read.]**
>
> We have updated our figure to make it clear and emphasized our methods with "ours" according to your suggestion.
>
> **[Q3. Figure 2. it's common to also add the parameters of the models in the plot ($\theta_y$, $\theta_t$, $\phi$). e.g. check also the original VAE paper.]**
>
> Thanks for this suggestion. We have added the parameters of the models in Figure 2 to make it clear.
>
> **[Q4. Figure 3. What does MiM stands for?]**
>
> MiM stands for the multiple index model. We have updated the full name of this abbreviation in the revision.
>
> **[Q5. Metrics: "we employ the widely recognized metric" - can you please provide citation?]**
>
> Thanks for your suggestion. We have added the related citations in the revision.
>
> **[Q6. "this metric quantifies the proximity of our predicted outcomes to the true potential outcomes when considering covariates and the corresponding treatment" -> from the definition of the metric this doesn't quantify the error for the corresponding treatment but for all possible treatments in T. Please update the text.]**
>
> Thanks for your advice. We have updated the text to *this metric quantifies the average approximating error of our predicted outcomes to the true potential outcomes for all possible treatments in $T$* in the revision.
>
> **[Q7. Figure 4. True response seems same between all plots, is this correct? So in that sense it's one simulation dataset but for different noise distributions?]**
>
> > Is this correct?
>
> Yes, this is correct.
>
> > it's one simulation dataset but for different noise distributions?
>
> In these four different simulations, we only change the noise distributions. We showcase these results to demonstrate the robustness of our methods in the face of diverse measurement error scenarios.
>
> The consistency across all true responses in the four sub-figures arises from the fact that they correspond to the same data generation process, but with varying types of measurement errors. It is essential to emphasize that the underlying true response remains unaffected by the measurement error, consistently maintaining its integrity across all sub-figures.
>
> **[Q8. Table 1. Why APE is 0.0 +- 0.0? The model seem to have high AMSE error but no error (!) on APE. Would be curious to understand what happened there.]**
>
> AMSE quantifies the error in outcome prediction, while APE measures the error in optimal policy estimation. While these two metrics are often correlated. We found that, in the News dataset, the optimal policy is relatively straightforward to learn as the optimal policy is the largest treatment and the true ATE increases with the treatment. However, the intricate nature of $p(T|\mathbf{X})$ contributes to a suboptimal performance in outcome prediction.
>
> **[Q9. Appendix B.4 last sentence. Fix the citation (?)]**
>
> We have fixed this citation issue in the revision.
>
> [1] [Hassanpour, Negar, and Russell Greiner. Learning disentangled representations for counterfactual regression. In International Conference on Learning Representations. 2019.](https://openreview.net/pdf?id=HkxBJT4YvB)
>
> [2] [Lizhen Nie, Mao Ye, Dan Nicolae, et al. Vcnet and functional targeted regularization for learning causal effects of continuous treatments. In International Conference on Learning Representations, 2020.](https://openreview.net/pdf?id=RmB-88r9dL)
>
> [3] [Khemakhem, Ilyes, Diederik Kingma, Ricardo Monti, and Aapo Hyvarinen. Variational autoencoders and nonlinear ica: A unifying framework. In International Conference on Artificial Intelligence and Statistics, pp. 2207-2217. PMLR, 2020.](https://proceedings.mlr.press/v108/khemakhem20a/khemakhem20a.pdf)
>
> [4] [Von Kügelgen, Julius, Yash Sharma, Luigi Gresele, Wieland Brendel, Bernhard Schölkopf, Michel Besserve, and Francesco Locatello. Self-supervised learning with data augmentations provably isolates content from style. Advances in neural information processing systems 34 (2021): 16451-16467.](https://openreview.net/pdf?id=4pf_pOo0Dt)

---

### Official Review · Reviewer_gZZT · 2023-11-07

**Soundness:** 3 good
**Presentation:** 4 excellent
**Contribution:** 3 good
**Rating:** 6
**Confidence:** 3

**Summary:**

The paper proposes a continous treatment effect estimation where treatment is observed with some measurement noise. The proposed algorithm builds upon variational auto-encoders and aim to maximize the ELBO objective. The proposed algorithm is then compared with state-of-art algorithms in synntetic and semi-synthetic datasets.

**Strengths:**

1. The paper is very well-written, easy for any reader to follow very easily. The problem is very-well motivated in the introduction.
2. To the best of my knowledge, the treatment measurement error is not investigated in the literature. This paper proposes a methodology to handle this case.
3. Using vational encoders in the treatment effect estimation is not a new idea, it is well-articulated in the paper.

**Weaknesses:**

1. I am not completely convinced about technical contribution of the paper. The treatment measurement error is the main contribution of the problem definition, but it is not clear what new challenges this brings. Is it a straightforward extension of the existing work ?
2. I think one of the weakest section in the paper is the experiments. The proposed algorithm is only compared with state-of-art algorithms in synthetic and semi-synthetic datasets where both treatments and outcomes are synthtically-generated. Additionally, IHDP data has only ~700 samples. It would have been much stronger if the proposed algorithms are tested in real-world dataset with more samples.

**Questions:**

1. What are the technical challenges that come with treatment measurement ?
2. What happens to performance of the algorithm if the treatments are measured  without an error ?

---

> ### Author Response · Authors · 2023-11-18
> **Response to reviewer gZZT (part 1/3)**
>
> We would like to thank the reviewer for the thoughtful comments and detailed comments. We provide point-wise responses to address your comments below. Notice that some of the discussions have been updated in the revision to further improve the quality of our paper.
>
> **[Q1. I am not completely convinced about technical contribution of the paper. The treatment measurement error is the main contribution of the problem definition, but it is not clear what new challenges this brings. Is it a straightforward extension of the existing work? What are the technical challenges that come with treatment measurement?]**
>
> > What are the technical challenges that come with treatment measurement?
>
> Thanks for your comments. With hindsight, we realized that we did not discuss clearly enough the challenge of the problem in the introductory section. However, we argue that achieving accurate treatment effects estimation (TEE) in the presence of measurement error on the treatment variable poses a non-trivial challenge.
>
> - Even within the linear regression framework, a direct regression of $Y$ on a noisy measurement of $X$ leads to attenuation error. This implies an underestimation of the regression coefficient due to the presence of measurement error (ME) [1]. This attenuation error would be similarly observed when estimating treatment effects, indicating that neglecting ME can **result in a biased model**. Therefore, how to establish an unbiased estimator for TEE is a crucial problem.
> - Moreover, the intricacy of the problem escalates when the ME occurs in the treatment variable. In existing methods for TEE, two key parts are undertaken to estimate the treatment effect: generalized propensity score (GPS) estimation, involving learning $\pi = p(T|\mathbf{X})$, and outcome prediction, involving learning $p(Y|T,\pi)$. As we can see, the influence of ME on the treatment is shown in both these aspects. This introduces notable bias due to the cumulative impact of errors in these two parts.
>
> The practical implications of this problem and the aforementioned challenges motivate the development of our proposed method.
>
> > Is your method a straightforward extension of the existing work?
>
> Thanks for your comments. We realized that we lacked a thoughtful discussion on the differences between our proposed method and existing works, which may raise your concern here.
>
> We are driven by the practical challenges when estimating TEE with measurement error on the treatment variable. To tackle this challenge, we take a novel Bayesian perspective, including data generation process modeling and maximizing the likelihood of estimating the model parameters. The key is that the likelihood involves an integral over the unmeasured treatment variable, which presents a challenge for analytical computation. We take a variational framework and aim to maximize the ELBO objective to address this problem.
>
> [Comparison with non-parametric method which deals with TEE with ME.] Huang and Zhang [2] tackled this challenge by identifying the average dose–response function (ADRF) through a weighted conditional expectation and constructing a deconvolution kernel estimator. However, their approach is **computationally intensive** (Fig. 3 in our paper) and encounters **scalability issues**, particularly when confronted with high-dimensional covariates. Our approach adopts a Bayesian perspective to formulate this problem, leveraging the maximization of marginal likelihood for estimating model parameters. This distinctive technique provides an alternative pathway to address the issue at hand. Importantly, our method excels in reducing computational complexity, offering a scalable solution that can accommodate higher-dimensional covariates.
>
> [Comparison with NN methods which ignore the ME.] As we discussed, ignoring the ME would lead to learning **a biased model** since the presence of attenuation error. Our objective is to design a model to learn an unbiased estimator of treatment effect. We harness the effectiveness of network-based methods within our framework, which is inherently network-agnostic. This implies the versatility to seamlessly integrate various base methods into our framework. In our paper, we illustrate this adaptability by employing examples such as varying coefficients network (VCNet) and attention-based network (ATTNet), underscoring the flexibility of our approach.
>
> In our revision, we have added a paragraph that introduces the challenges that come with treatment measurement in the introduction section.

---

> ### Author Response · Authors · 2023-11-18
> **Response to reviewer gZZT (part 2/3)**
>
> **[Q2. I think one of the weakest section in the paper is the experiments. The proposed algorithm is only compared with state-of-art algorithms in synthetic and semi-synthetic datasets where both treatments and outcomes are synthetically generated. Additionally, IHDP data has only 700 samples. It would have been much stronger if the proposed algorithms are tested in real-world dataset with more samples.]**
>
> > Why do we test all methods on simulations and semi-synthetic data?
>
> The primary challenge in causal inference lies in the unobservability of counterfactual outcomes. In other words, the true treatment effect is never observed (i.e., missing in any training data), rendering it impractical for training predictive models. This makes estimating treatment effects a more difficult problem than the standard supervised learning problem. Furthermore, the challenge extends to not allowing for meaningful evaluation of real-world datasets due to the inability to observe the counterfactuals. Consequently, akin to prior research practices [3,4,5,7], we resort to employing well-established benchmark synthetic models to assess the performance of all treatment effect estimation methods.
>
> > Large dataset with more samples.
>
> Thanks for your suggestion. We have expanded our analysis to include experimental results from a larger dataset, namely The Cancer Genomic Atlas (TCGA), which encompasses gene expression measurements for cancer patients [6]. There are in total $9659$ samples for which we used the measurements from the $4000$ most variable genes. In alignment with established methods [5,7], we have adhered to the same procedures for generating the values treatment and outcome variables. Details of this comparison are presented in the following table.
>
> | Method          | TCGA (AMSE)      | TCGA (APE)      |
> |-----------------|------------------|-----------------|
> | TARNet          | 0.042 ± 0.007    | 0.136 ± 0.069   |
> | DRNet           | 0.047 ± 0.007    | 0.182 ± 0.040   |
> | ATTNet          | 0.056 ± 0.019    | 0.076 ± 0.050   |
> | VCNet           | 0.046 ± 0.010    | 0.067 ± 0.045   |
> | VF-ATTNet(Ours) | 0.042 ± 0.010    | 0.057 ± 0.037   |
> | VF-VCNet (Ours) |**0.019 ± 0.014** |**0.036 ± 0.017**|
>
> Compared to the baseline methods, our approaches consistently demonstrate enhanced performance across both AMSE and APE metrics. Notice that our two methods should be compared with their respective base methods, namely VF-VCNet versus VCNet and VF-ATTNet versus ATTNet.
>
> We have updated this result in the revision to further verify the effectiveness of our proposed method.

---

> ### Author Response · Authors · 2023-11-18
> **Response to reviewer gZZT (part 3/3)**
>
> **[Q3. What happens to performance of the algorithm if the treatments are measured without an error?]**
>
> Notice that in our paper, we develop our method under the assumption that **the distribution formulation of the measurement error, denoted as $U$, is known.** When there is no measurement error present, the distribution of the measurement error becomes a discrete distribution with its probability mass concentrated solely on zero. Consequently, it allows for the direct determination of the true variable $T$ without the need for a variational step to infer $T$ from observed variables $S$ and $Y$. **In this case, our variational framework would degenerate to the basic model that directly takes $T$ to estimate the treatment effect.**
>
> In real-world scenarios [1], researchers usually devise specific methods to deal with measurement errors in situations where there is a confirmed presence of measurement error in the data. This suspicion typically arises when there are inconsistent multiple measurements of a variable for a single individual. Furthermore, the existence of multiple measurements enables the identification of characteristic functions associated with the measurement error [2,8]. Building upon this fact, our proposed method capitalizes on prior knowledge of the measurement error and strategically maximizes the joint likelihood to get an unbiased estimator for treatment effects.
>
> We hope this clarification addresses your question adequately. Additionally, we have incorporated this discussion into the Appendix to help any audience who may have concerns on this point.
>
> [1] [Carroll R. J., Rupper D., Stefanski L. A., Crainiceanu C. M. (2006). Measurement error in nonlinear models: A modern perspective. Chapman Hall/CRC.](https://www.taylorfrancis.com/books/mono/10.1201/9781420010138/measurement-error-nonlinear-models-ciprian-crainiceanu-raymond-carroll-leonard-stefanski-david-ruppert)
>
> [2] [Huang, W. and Zhang, Z. (2023). Nonparametric estimation of continuous treatment effect with measurement error. JRSSB, 85, 474 - 496.](https://academic.oup.com/jrsssb/article/85/2/474/7098034)
>
> [3] [Claudia Shi, David Blei, and Victor Veitch. Adapting neural networks for the estimation of treatment effects. In Advances in Neural Information Processing Systems, pp. 2503–2513, 2019.](https://proceedings.neurips.cc/paper/2019/file/8fb5f8be2aa9d6c64a04e3ab9f63feee-Paper.pdf)
>
> [4] [Lizhen Nie, Mao Ye, Dan Nicolae, et al. Vcnet and functional targeted regularization for learning causal effects of continuous treatments. In International Conference on Learning Representations, 2020.](https://openreview.net/pdf?id=RmB-88r9dL)
>
> [5] [Xin Wang, Shengfei Lyu, Xingyu Wu, Tianhao Wu, and Huanhuan Chen. Generalization bounds for estimating causal effects of continuous treatments. Advances in Neural Information Processing Systems, 35:8605–8617, 2022.](https://proceedings.neurips.cc/paper_files/paper/2022/file/390bb66a088d37f62ee9fb779c5953c2-Paper-Conference.pdf)
>
> [6] [John N Weinstein, Eric A Collisson, Gordon B Mills, Kenna R Mills Shaw, Brad A Ozenberger, Kyle Ellrott, Ilya Shmulevich, Chris Sander, Joshua M Stuart, Cancer Genome Atlas Research Network, et al. The cancer genome atlas pan-cancer analysis project. Nature Genetics, 45(10):1113, 2013.](https://www.cancer.gov/ccg/research/genome-sequencing/tcga)
>
> [7] [Bica, Ioana, James Jordon, and Mihaela van der Schaar. Estimating the effects of continuous-valued interventions using generative adversarial networks. Advances in Neural Information Processing Systems 33 (2020): 16434-16445.](https://proceedings.neurips.cc/paper/2020/file/bea5955b308361a1b07bc55042e25e54-Paper.pdf)
>
> [8] [Delaigle, Aurore, Jianqing Fan, and Raymond J. Carroll. A design-adaptive local polynomial estimator for the errors-in-variables problem. Journal of the American Statistical Association 104.485 (2009): 348-359.](https://www.ncbi.nlm.nih.gov/pmc/articles/PMC2846380/)

---

### Meta-Review · Area_Chair_qAgZ · 2023-12-05

**Metareview:**

The authors study estimating treatment effects in the observation setting with a twist: the treatment is not directly observed, but is rather contaminated with some measurement noise. They treat the treatment as a latent variable and propose an approach using variational auto-encoders to learn the treatment effects. The authors have done adequate simulations and the reviewers appreciated the novel formulation of the problem (i.e., considering measurement noise in the model).

**Justification For Why Not Higher Score:**

I am not quite as excited as the reviewers about the results. It is combines various standard approached in Bayesian analysis and contains no theoretical results.

**Justification For Why Not Lower Score:**

The reviewers liked the novelty of incorporating noise into the treatment model and the authors did a good job engaging with the reviewers.

---

### Decision · Program_Chairs · 2024-01-16

Accept (poster)